# Fully differentiable, fully distributed Rainfall-Runoff Modeling

Fedor Scholz<sup>1</sup>, Manuel Traub<sup>1</sup>, Christiane Zarfl<sup>2</sup>, Thomas Scholten<sup>3</sup>, and Martin V. Butz<sup>1</sup>

**Correspondence:** Fedor Scholz (fedor.scholz@uni-tuebingen.de)

**Abstract.** Traditional hydrological modeling simulates rainfall-runoff process dynamics using process-based models (PBMs). PBMs are grounded in physical laws and therefore highly interpretable. As environmental systems are highly complex, though, subprocesses are sometimes hard or even impossible to identify and quantify. Data-driven approaches, like artificial neural networks (ANNs), offer an alternative. Such approaches can automatically discover hidden relationships within the data. As a result, superior model performance may be achieved. However, the uncovered relationships are hard to analyze within blackbox ANNs and often fail to respect physical laws. Differentiable modeling calls for knowledge discovery by combining both approaches, benefiting from their respective advantages. In this work, we present a physically inspired, fully differentiable, and fully distributed model, which we term DRRAiNN (Distributed Rainfall-Runoff ArtIficial Neural Network), DRRAiNN is a neural network model that estimates river discharge based on meteorological forcings and elevation. Focusing on the Neckar river catchment in Southwest Germany, DRRAiNN is trained to predict daily water discharge measurements using data from 17 stations and from ten meteorological years only. DRRAiNN's performance is compared to the performance of the European Flood Awareness System (EFAS) reanalysis. Some instances of our model outperform EFAS at lead times of over 50 days in terms of the applied metrics for model performance. As DRRAiNN is fully differentiable and fully distributed, efficient source allocation algorithms can be used to identify the precipitation sources responsible for the water discharge dynamics at specific gauging stations. Besides DRRAiNN's potential to forecast upcoming water discharge dynamics, its full differentiability could be utilized to infer erosion sites from turbidity data, particularly when integrated with an appropriate erosion model.

#### 1 Introduction

Accurate water flow forecasting plays a critical role in mitigating short-term flood impacts, such as preventing loss of life and reducing economic damage (Pilon, 2002). For example, simulating river discharge is a prerequisite for flood inundation modeling (Hunter et al., 2007) and enables informed decision-making in water management such as dam operations (Valeriano et al., 2010). Accuracy is not everything though. Hydrological models that respect physical laws are more likely to generalize well to new situations and to offer insights into the underlying processes that govern water movement. A solid understanding of the dynamics of water systems is necessary to estimate the impacts of environmental planning and to improve infrastructure design (Palmer et al., 2008; Bharati et al., 2011). It also enables a better assessment of how climate change may alter existing ecosystems in the future (Palmer et al., 2008; Van Vliet et al., 2013; Al Hossain et al., 2015). Additionally, models that respect

<sup>&</sup>lt;sup>1</sup>Neuro-Cognitive Modeling Group, University of Tübingen, Tübingen, Germany

<sup>&</sup>lt;sup>2</sup>Environmental Systems Analysis, University of Tübingen, Tübingen, Germany

<sup>&</sup>lt;sup>3</sup>Soil Science and Geomorphology, University of Tübingen, Tübingen, Germany

physical laws can be used to infer the origins of observed discharge, thereby further facilitating the development of policies that mitigate the damages caused by floods. From a practical perspective, a good model should allow efficient calibration and perform well even if data are sparse, which is often the case for river discharge. Traditionally, these challenges have been addressed using physically based approaches that explicitly encode domain knowledge. These process-based models (PBMs) describe physical processes with mathematical equations derived from physical laws and observations (Brutsaert, 2023).

Environmental hydrological processes are highly complex, involving numerous interacting variables that make the overall process highly heterogeneous (Marçais and de Dreuzy, 2017). Recent advances, such as the Multiscale Parameter Regionalization framework (Samaniego et al., 2010) and scalable transfer function approaches (Imhoff et al., 2020) have focused on improving parameterization and capturing spatial heterogeneity in PBMs to alleviate these issues. To reduce uncertainty and initialize PBMs adequately, data assimilation incorporates concrete observations into running models (Liu et al., 2012; Camporese and Girotto, 2022; Montzka et al., 2012). Such advancements in data assimilation can improve performance in both lumped (Moradkhani et al., 2005; Liu and Gupta, 2007; Liu et al., 2012) and distributed models (Rakovec et al., 2012). However, significant challenges remain, as the involved processes and their interactions are in most cases only partially understood (Hrachowitz et al., 2013), leading to high uncertainty and biases Even if a process is known well in detail, certain input variables may simply be unobservable, such as underground topography. Additionally, environmental processes often occur at scales that differ substantially from those observed under laboratory conditions (Hrachowitz et al., 2013; Shen, 2018; Nearing et al., 2021).

Complementary to PBMs, data-driven models have gained traction in recent years, driven by the increasing amount of available hydrological data (Sit et al., 2020). Artificial neural networks (ANNs) are data-driven models that automatically learn relationships from large datasets. Given the superior performance of early data-driven approaches in hydrology, it is likely that the full potential of data-driven approaches remains untapped (Shen, 2018; Nearing et al., 2021). However, despite achieving strong predictive performance, ANNs often fail to respect physical laws due to their purely data-driven nature. This calls for measures such as hybrid or physics-informed models that bias data-driven approaches toward physical plausibility. Furthermore, it is often criticized that developers of machine learning (ML) models do not put enough effort into the interpretation of their developed systems, failing to gain a better understanding of the system's internal dynamics (Muñoz-Carpena et al., 2023).

One promising avenue to overcome these limitations involves leveraging ML to infer latent variables that are otherwise inaccessible to direct measurement. To give an example, a considerable portion of total discharge originates from subsurface flow. It is not yet possible to directly measure subsurface flow, making underground topography a latent driver of hydrological behavior (Shen, 2018). We believe that these latent variables may contribute to poor model generalization across basins. ML and especially ANNs can support hydrological modeling in such cases, because they allow to infer latent variables retrospectively given observation dynamics (Butz et al., 2019; Otte et al., 2020). This motivates a key question we address in this paper: *Given observed dynamics, in which areas did precipitation contribute to the measured discharge?* 

Similar to subsurface flow, evapotranspiration cannot be directly measured and must also be inferred indirectly. Model inversions of NNs (Sit et al., 2020) may therefore help to extend our understanding of the water cycle with ML. For a broader overview of ML applications in hydrology, we refer the reader to Shen (2018) and Sit et al. (2020).

A combination of PBMs and ML-based approaches could leverage the advantages of both worlds. When combined with the goal of knowledge discovery, this approach is referred to as "differentiable modeling" (Shen et al., 2023). It could result in well-performing interpretable models that automatically find new relationships in the data, respect physical laws, generalize well across different settings, and require comparatively little data. From the ML perspective, known relationships can be incorporated into differentiable models as constraints or inductive biases. Inductive biases encode prior assumptions about the data-generating process, effectively constraining the model's solution space. By doing so, they can improve performance, enhance generalization, and make learning more efficient. Furthermore, they guide the model towards discovering interpretable structures in the data, aligning its behavior with established principles (Butz et al., 2024). A crucial challenge for the modeler is to find and incorporate those biases that restrict the solution space as much as possible without introducing incorrect or unjustified assumptions and without restricting the self-organizing power of NNs.

In their seminal work, Kratzert et al. have successfully used a long short-term memory (LSTM) (Hochreiter and Schmidhuber, 1997) for rainfall-runoff modeling at the basin scale (Kratzert et al., 2018), demonstrating that purely data-driven models can exceed traditional methods. Since then, numerous studies have emerged, applying largely the same model to various data sets (Sit et al., 2020). Notably, significant advancements to the model have also been made, including the incorporation of physical constraints (Kratzert et al., 2019; Hoedt et al., 2021), uncertainty estimation (Klotz et al., 2022), and the extension of modeling to multiple timescales (Gauch et al., 2021). Hybrid models such as neural ODEs, where differential equations of conceptual hydrological models are replaced by neural networks, were also applied in this setting (Höge et al., 2022). All of the aforementioned models are lumped, meaning that inputs are spatially aggregated over each catchment. These catchments are typically delineated using digital elevation models.

Semi-distributed models partially leverage river network topology, providing a compromise between lumped and fully distributed representations. These include purely data-driven graph-based models (Xiang and Demir, 2020; Moshe et al., 2020; Sit et al., 2021; Kratzert et al., 2021; Sun et al., 2022; Chen et al., 2022), as well as hybrid approaches that integrate domain knowledge – for example, by using a differentiable Muskingum-Cunge routing model (Bindas et al., 2024; Zhong et al., 2024). These models typically divide the overall catchment into multiple subbasins connected via the river network, enabling limited spatial interaction. Within each subbasin, however, forcings are still spatially aggregated, similar to lumped models.


In contrast, fully distributed models directly operate on a spatial grid. While there is a call for more fully distributed data-driven models for rainfall-runoff modeling (Nearing et al., 2021), most existing approaches remain limited in critical ways. Some hybrid models operate on a grid but restrict cell-to-cell communication to the direction of steepest descent (Xiang and Demir, 2022; Wang et al., 2024). This strong assumption effectively transforms the grid into a directed graph, excluding physically plausible underground flows in other directions. CNN-LSTMs process gridded input data without explicit assumptions about flow directions (Ueda et al., 2024; Pokharel and Roy, 2024a; Li et al., 2022). However, these models separate spatial and temporal processing by flattening the convolutional neural network (CNN) outputs before passing them to an LSTM. As a result, spatial dependencies are not maintained across time steps. This limitation is addressed in Oddo et al. (2024), were a ConvLSTM (Shi et al., 2015) is used to jointly model space and time. Yet, before the final discharge prediction, the outputs of all grid cells are flattened into a single feature vector and passed through a fully connected layer. Similar global aggregation

strategies can be found elsewhere (Zhu et al., 2023; Tyson et al., 2023; Pokharel and Roy, 2024b; Xu et al., 2022; Börgel et al., 2025). Moving a step closer to physical plausibility, Longyang et al. (2024) combined a ConvLSTM with ridge regression to learn which grid cells should contribute to discharge estimation at each station. This allowed the reconstruction of plausible underground flow paths between subbasins. Since all of these distributed models aggregate the outputs of the spatial component globally over space, whether weighted or not, they lack the incentive to propagate water across the landscape in a physically plausible way.

Our work builds on differentiable modeling to combine both process-based and data-based modeling, and to address the challenges of physical plausibility, interpretability, and latent variable inference. We present DRRAiNN (Distributed Rainfall-Runoff ArtIficial Neural Network), a physics-inspired, fully differentiable, fully distributed rainfall-runoff model. Our spatio-temporal ANN architecture estimates river discharge at gauging stations from gridded precipitation, solar radiation, elevation, and past discharge. DRRAiNN is fully distributed in the sense that it internally operates on a grid. However, its outputs are point-wise river discharge measurements at given gauging station locations. Its full differentiability allows gradients to flow seamlessly through the entire system, enabling end-to-end optimization of all its components with sparse discharge measurements being the only target variable. To avoid overfitting, and to improve interpretability and generalization, we incorporated several physics-inspired inductive biases into DRRAiNN. These include the modularization into a spatially fully distributed rainfall-runoff model and the utilization of a graph-based river discharge model. Additional architectural choices precondition DRRAiNN to encode distinct processes, such as lateral propagation of water across the landscape and local evapotranspiration. As a result, DRRAiNN turns into a gray-box deep learning model. Its model design encourages the development of sub-modules, which model surface and sub-surface water flow, water inflow into a river network, and water flow and discharge across the river network.

Thanks to DRRAiNN's fully distributed and fully differentiable architecture, it is possible to answer spatially resolved questions, such as: Where is the true catchment area, including contributions from underground flows? In other words, DRRAiNN enables source allocations using gradient-based attribution methods like integrated gradients (Sundararajan et al., 2017). These techniques can help to examine and understand internal model dynamics, enabling knowledge discovery.

## 120 2 Methods




We present DRRAiNN, a spatio-temporal ANN architecture that estimates river discharge from static attributes and meteorological forcings in a distributed manner. We evaluate DRRAiNN's estimation abilities, physical plausibility, and the necessity of its architectural design choices. We demonstrate its performance in a real-world setting on the Neckar River in Southwest Germany, comparing it to simulations from the European Flood Awareness System (EFAS, Mazzetti et al. (2023)). DRRAiNN achieves higher KGE and NSE values than EFAS for lead times of up to 50 days and provides interpretable source attributions that enable the reconstruction of effective catchment areas from modeled dynamics.

## 2.1 Model



DRRAiNN's structure is grounded in the following data and structural information sources. The locations  $L_i = (x_i, y_i)$  for estimations of discharge in the river network are determined by discharge gauging stations that provide observed discharge  $Q_{i,t}$  for time t in 24 h periods. The connectivity of stations, determined by the river network, is encoded in an adjacency matrix  $A_{i,j}$ . Static maps  $S_{x,y}$  and meteorological forcings  $F_{x,y,t}$  for hourly time points t are encoded on a grid that spans the whole catchment area of the river network. Given static maps  $S_{:,:}$ , meteorological forcings  $F_{:,:,t_0:t_s+T}$  over the whole duration  $(t_0 \dots t_s + T)$  in hours, and past discharge  $Q_{i,t_0:t_s}$  over the tune-in period  $(t_0 \dots t_s)$  in days, DRRAiNN estimates discharge  $Q_{i,t_s+1:t_s+T}$  over a temporal future horizon of T days via a function f, representing the learned spatio-temporal mapping implemented by the model:

$$\tilde{Q}_{i,t_0+1:t_0+T} = f(S_{:::}, F_{::::t_0:t_0+T}, Q_{i,t_0:t_0}) \tag{1}$$

Since surface and subsurface flow differ from river flow dynamics as described above, we model these subprocesses separately. Therefore, DRRAiNN consists of two components, the rainfall-runoff model and the discharge model. The rainfall-runoff model operates recurrently on a grid, rendering it fully distributed. It is supposed to model surface and subsurface flow, and evapotranspiration. The discharge model operates recurrently on a graph to model river flow inside of channels and output estimated discharge  $\tilde{Q}$  at the station locations. While DRRAiNN is fully distributed in its internal computation over a spatial grid, its outputs are only available at selected gauging stations.

At each time step, DRRAiNN processes the sequence in an auto-regressive loop by first invoking the rainfall-runoff model, followed by the discharge model. The rainfall-runoff model receives gridded static maps S and meteorological forcings F as input to model the catchment on a grid. It is primed to distinguish between two important subprocesses, namely surface and subsurface flow, which is mainly driven by topography, and evapotranspiration, which is mainly driven by solar radiation. It produces a latent representation, which we term runoff embedding, extracted at station locations and used as input to the discharge model. Despite being the main driver of discharge, it cannot be directly interpreted as runoff due to its self-organizing nature. The discharge model additionally receives an adjacency matrix A that describes the connectivity between stations, static river segment features, and the (potentially estimated) discharge  $Q_{:,t-1}$  from the previous time step. It then estimates discharge  $\tilde{Q}$  for each station, from which the training loss is computed.

We implement DRRAiNN in pytorch (Paszke et al., 2019). In the following, we provide a more detailed description of DRRAiNN's components. See Fig. 1 for a depiction of the overall model.

## 2.1.1 Rainfall-Runoff Model

The rainfall-runoff model consists of a position-wise LSTM and a CNN that are called in each time step. This renders the rainfall-runoff model local in space and time. Only spatially local and temporally previous information is used to update internal states.

Figure 1. Schematic overview of the DRRAiNN architecture. The gridded rainfall-runoff model has two main tasks: to model the redistribution of precipitation across the landscape, and to model evapotranspiration based on solar radiation. It receives precipitation as its main input to a point-wise LSTM, whose hidden states, but not cell states, are updated using a ConvNeXtBlock. The ConvNeXtBlock weights are not fixed but dynamically generated by hypernetworks (indicated by red arrows). The depth-wise convolution (DWConv), responsible for lateral water propagation, receives its weights from a CNN that takes elevation as input and shares the same receptive field as the DWConv. The point-wise convolutions (PWConv1 and PWConv2), used to model local evapotranspiration processes, receive their weights from an MLP that takes solar radiation as input. The LSTM hidden state is further processed by a linear layer before being passed to the discharge model. This graph-based discharge model aggregates information at the gauging stations, incorporating the last (possibly inferred) discharge values, elevation differences between stations, and river segment lengths. Its output is the estimated discharge at each station.

## 2.1.2 Modeling temporal dynamics






The position-wise LSTM (PWLSTM) is responsible for modeling the temporal relationships in the data and therefore maintains a hidden and a cell state for each grid cell. The gating mechanism regulates when and how the cell state is updated, allowing the model to retain information over extended time periods. This can be particularly useful for implicitly modeling slow hydrological processes such as soil moisture or groundwater levels, which evolve more gradually than overland flow. The LSTM receives precipitation as input to update its hidden and cell states. It has a hidden size of 4 (see Appendix B for hidden sizes 2 and 6). Importantly, the weights of the LSTM are shared throughout the gridded area. As a result, while the LSTM at each grid cell maintains individual hidden and cell state values, the temporal processing principle is identical everywhere. The assumption is that the unfolding physics is the same everywhere, although they may be locally parameterized.

# 2.1.3 Modeling spatial dynamics

The CNN models spatial relationships such as the propagation of water flow across the landscape and evapotranspiration. It receives and updates the hidden state h of the PWLSTM to model spatial interactions, while leaving the PWLSTM's cell states untouched to preserve temporal memory. Surface and subsurface flow are spatially extended processes, whereas evapotranspiration is primarily a local phenomenon, occurring independently at each grid cell. To reflect this distinction, we separate the CNN's treatment of these processes using different convolution types and input sources, introducing an inductive bias into the architecture.

More precisely, the CNN is based on a modified ConvNeXt block (Liu et al., 2022). A ConvNeXt block consists of three layers, namely a depth-wise convolutional layer (DWConv) with kernel size  $7 \times 7$  followed by a position-wise inverted bottleneck given by two linear layers (PWConv1 and PWConv2). This way, ConvNeXt decouples spatial and channel-wise information flow. We apply the SiLU activation function after the convolutional and between the linear layers (Hendrycks and Gimpel, 2016). In contrast to its original formulation, the weights of our ConvNeXt block are not static but location-dependent. They are parameterized by other neural networks, turning this network component into a *hypernetwork* (Traub et al., 2024). This means that the ConvNeXt block can behave differently at each location on the grid. Calling DWConv results in the following operation:

$$y_{i,j,c} = \sum_{m=-3}^{3} \sum_{n=-3}^{3} w_{i,j,m,n,c} \cdot x_{i+m,j+n,c},$$
(2)

where y is the output, x the input, w are the weights produced by the hypernetwork, c is the considered channel, and i and j are coordinates. We can still call this operation a convolution if we regard the input variables together with the weight-generating networks as the kernel. Calling PWConv1 and PWConv2 results in the following operation:

$$y_{i,j,c_{\text{out}}} = \sum_{c_{\text{in}}} w_{i,j,c_{\text{out}},c_{\text{in}}} \cdot x_{i,j,c_{\text{in}}},\tag{3}$$

Each layer of the ConvNeXt block is parameterized by a distinct hypernetwork, tailored to the type of process it represents. The weights of DWConv are produced by a CNN that has the same kernel size as DWConv itself. The weights for PWConv1 and

**Figure 2.** Illustration of the hypernetworks used in DRRAiNN. In both panels, the dark gray cells represent locations whose hidden states are updated based on information from the light gray cells. The weights for these updates are generated by separate neural networks that share the same receptive field but receive different types of input data. Left: A CNN takes elevation as input and produces the weights for the depth-wise convolution, which models lateral water propagation. Right: An MLP takes solar radiation as input and produces the weights for the point-wise convolution, which models localized evapotranspiration.

PWConv2 are produced by position-wise multi-layer perceptions (MLPs). By using different input variables for the different hypernetworks, we can distinguish between local and spatially extended processes. How water propagates across the landscape depends mainly on the topography, which is why we generate the weights of DWConv from elevation. Before feeding the elevation into the hypernetwork, we subtract the elevation of the center cell from the elevations of all other cells within each receptive field as relative elevation is more informative for flow direction than absolute elevation. Evapotranspiration, on the other hand, is a local process and is therefore best captured by the position-wise components. This is why we generate the weights for PWConv1 and PWConv2 from solar radiation. See Fig. 2 for an illustration.

## 2.1.4 Adapter

Lastly, the runoff embeddings are extracted at the station locations, fed through a single linear layer, and sent to the river discharge model. Aggregating the hidden states of all cells on the corresponding upstream river segment showed a tendency to overfit in preliminary experiments.

## 200 2.1.5 Discharge Model


Our discharge model is a recurrent graph neural network called DISTANA (Karlbauer et al., 2019), with the graph structure defined by the actual river network and the stations. DISTANA maintains two types of recurrent units: station and segment kernels, both implemented as Gated Recurrent Units (GRUs, Cho et al. (2014)) with a hidden size of 8 (see Appendix B for hidden sizes 4 and 16, and a version in which the GRUs are replaced with LSTMs). Station kernels are placed at the gauging stations, while segment kernels are located on segments between stations. These kernels communicate with each other via

lateral connections with 4 channels (Fig. 1). In each time step, the segment kernels are updated first, followed by the station kernels, which then estimate the discharge  $\tilde{Q}$  at their respective locations. The segment kernels first concatenate the previous output of the upstream station kernels with static river segment attributes – specifically the altitude difference and segment length. After applying the GRU, the output is multiplied by the adjacency matrix, which is derived from the river network topology and station positions. The segment kernels thereby sum up information from upstream station kernels. The output of the segment kernels serves as input for the station kernels. The station kernels work similarly. They first concatenate the last output of the segment kernels with the last (potentially inferred) discharge and the output of the rainfall-runoff model. After applying the GRU, the output is split into the estimated discharge  $\tilde{Q}$  and the input for the segment kernels in the next time step.

Although DRRAiNN receives hourly meteorological forcings F, it produces discharge estimates at a daily resolution. During the initial 10 day tune-in phase of each sequence, we feed the same observed discharge value Q into DRRAiNN for each hourly step within the day.

#### 2.2 Data




The input data for DRRAiNN consists of radar-based precipitation, elevation for above-ground topography, solar radiation, and river discharge data. Preliminary experiments showed no improvement when including temperature; therefore, we exclude it following Occam's razor.

For precipitation, we use the radar-based precipitation product RADOLAN provided by the Deutsche Wetterdienst (RADOLAN, 2016). The data domain is a  $900 \text{ km} \times 900 \text{ km}$  pixel grid with a resolution of  $1 \text{ km} \times 1 \text{ km}$  that covers all of Germany and a temporal resolution of 1 h. This grid defines the spatial resolution at which our model operates. RADOLAN data is log-standardized before being sent to the model due to its long-tail distribution. Specifically, we add 1 and take the logarithm, then compute the mean and standard deviation of the transformed data to standardize it. We replace missing values with zeros, which is the standardized mean.

For static topography information we use the digital elevation model (DEM) EU-DEM v1.1 provided by the Copernicus Land Monitoring Service of the European Environment Agency (EU-DEM, 2016). We also use the DEM to compute the differences in altitudes between adjacent discharge gauging stations. Elevation values and derived difference are standardized before being sent to the model, i.e., we subtract their mean and divide by their standard deviation.

For solar radiation, we use surface short-wave downward radiation (SSRD) from the ERA5 data set (Hersbach et al., 2018). It comes with a temporal resolution of 1 h and a relatively coarse spatial resolution of  $0.25^{\circ} \times 0.25^{\circ}$ . Like the precipitation data, solar radiation data is log-standardized. We use rasterio (Gillies and others, 2013) to transform and reproject the DEM and solar radiation data to match the RADOLAN coordinate reference system.

The topography of our river network is determined by the AWGN data set (AWGN, 2023). We use it to compute the adjacency matrix that describes which stations are connected via river segments and the corresponding river segment lengths.

Finally, we use discharge measurement data to tune in the discharge model and, more importantly, as the only target variable to train, validate, and test our model. We use data collected and provided by the German Federal Institute of Hydrology via the Global Runoff Data Centre (GRDC, 2024). The data set contains observed daily river discharge from gauging stations

worldwide, including those in Germany. Since the location information of the discharge gauging stations is partially wrong, we corrected them manually. We then align the station locations to the nearest river segment (snapping). If the correction exceeds a predefined threshold, the station is excluded. If two stations are very close to each other, one of them is discarded. Due to its long-tail distribution, discharge data is log-standardized on a per-station basis before being sent to the model. We add 1 and take the logarithm, then standardize the data using station-wise means and standard deviations. We replace missing values with zeros, which is the standardized mean of the corresponding station.

Our choice of input datasets was guided by temporal resolution, data provenance, and practical availability. Although the European Flood Awareness System (EFAS) employs EMO-1 for precipitation input, we opted for RADOLAN due to important differences: EMO-1 offers a coarser 6 hresolution and is interpolated from sparse station data, in contrast to RADOLAN's direct radar-based observations. Although we expect only minor differences in performance in some settings, radar-derived datasets like RADOLAN provide finer spatial and temporal resolution, which is advantageous for distributed models. Similarly, we chose ERA5 for solar radiation data due to its gridded format and hourly resolution. Alternative datasets, such as those provided by DWD, are either available only as station-wise hourly data, which lack the required grid format, or as gridded data aggregated monthly, which does not meet our temporal requirements. Daily datasets like EOBS may suffice if subdaily temporal patterns are encoded separately, but this would require additional preprocessing. A transition toward operation flood forecast would place increased importance on the choice of precipitation forecast products (Imhoff et al., 2022). Ultimately, all data products entail inherent uncertainties and errors, and our choices reflect a balance between data availability, temporal resolution, and the specific requirements of our model.

# 2.3 Study site



The Neckar river network in Southwest Germany spans a catchment area of  $14~000~\mathrm{km}^2$  with a mean elevation of  $460~\mathrm{m}$ . According to ERA5, temperatures in this region ranged from  $-25~\mathrm{^{\circ}C}$  to  $40~\mathrm{^{\circ}C}$  during our training period. Our dataset includes measurements from  $17~\mathrm{gauging}$  stations distributed across the river network (see Fig. 3). At the most downstream station in Rockenau, discharge during the training period ranged from  $29.5~\mathrm{m}^3/\mathrm{s}$  to  $1690~\mathrm{m}^3/\mathrm{s}$  with a mean of  $133.3~\mathrm{m}^3/\mathrm{s}$ .

The catchment features a highly heterogeneous landscape, including narrow and wide valleys, diverse geology (e.g., lime-stone, sandstone), different soil textures (e.g., clay, marl), and subsurface structures such as karst systems and pore water aquifers. This makes the modeling of the Neckar River network a challenging endeavor. To give a concrete example, there are underground flows south of Pforzheim that route water toward the east, while the elevation model suggests a different flow direction. (Ufrecht, 2002). This relationship cannot be inferred from a digital elevation model alone. Latent underground structures route the water in a different direction than the elevation model alone would suggest.

By restricting the domain to the Neckar river network, we end up with an area of size  $200 \text{ km} \times 200 \text{ km}$ . Following the transformations described above, all gridded data is reduced from a  $1 \text{ km} \times 1 \text{ km}$  grid to a  $4 \text{ km} \times 4 \text{ km}$  grid by taking the mean. This results in a  $50 \times 50$  grid covering the study area. We train our model on hydrological years 2006 - 2015, validate on 2016 - 2018, and test on 2019. Forcings F are provided at hourly resolution, while discharge is provided at daily resolution.

Figure 3. The study area used in this work is the Neckar River catchment in Southwest Germany.

**Table 1.** Truncation length schedule in days for TBPTT

| #Epochs | Truncation length | Batch size |
|---------|-------------------|------------|
| 10      | 1                 | 256        |
| 4       | 2                 | 128        |
| 2       | 4                 | 64         |
| 1       | 10                | 32         |
| 1       | 20                | 32         |

## 2.4 Experimental setup






We train DRRAiNN on sequences of 20 days (480 hourly steps), using the first 10 days as a warm-up phase. During this phase, we feed the model observed discharge values to initialize and align its hidden states with the true system dynamics. This procedure resembles data assimilation in traditional hydrological models, where observations are used to update model states and reduce uncertainty. In ML terms, this corresponds to teacher forcing. The warm-up phase allows the rainfall-runoff component of DRRAiNN to infer latent hydrological states, such as soil moisture or aquifer recharge, through its hidden state representations. This alignment helps the model transition smoothly to predictive, open-loop mode, where future discharge is estimated without access to ground-truth values.

After the warm-up phase, DRRAiNN transitions into open-loop mode for the remaining 10 days of each sequence. In this predictive mode, the discharge model feeds its own previous discharge estimations as inputs for subsequent time steps. The rainfall-runoff model, in contrast, continues to receive observed precipitation and solar radiation as inputs throughout the sequence. While informative, this setup does not reflect realistic operational conditions for discharge forecasting. Precipitation forecasting, in particular, remains a major challenge. Currently no algorithm can accurately predict precipitation 10 days ahead at a spatial resolution of  $4 \text{ km} \times 4 \text{ km}$ . However, this setup is well suited for knowledge discovery concerning hydrologic processes, which is primary focus in this work. We leave the evaluation of DRRAiNN under realistic, forecast-based conditions for future work.

We use the mean squared error (MSE) computed on station-wise standardized discharge data as both the training and validation loss. Standardization ensures that stations with larger discharge values do not dominate the loss, promoting a balanced learning across all stations. Training is performed using truncated backpropagation through time (TBPTT), where the truncation length increases progressively over the course of training. Initially, we backpropagate the loss over 1 day sequences (24 time steps) to help DRRAiNN focus on short-term temporal relationships and stabilize learning. Over the course of training, we increase the truncation length, enabling the model to learn longer-term dependencies. The truncation length schedule is shown in Table 1. We adapt the batch size to fit the model within the memory constraints of a single NVIDIA A100 GPU, with total training time remaining under 8 h. A forward simulation of a 20 day sequence takes approximately 4 s.

To improve generalization and account for model variability due to random initialization, we train five independent instances of DRRAiNN per experiment, each initialized with a different seed. We report test results based on the three runs with the

lowest validation loss out of five seeds. This selection procedure is applied consistently to both the primary model and all ablation variants. We use the Ranger optimizer (Wright, 2019) with a learning rate of 0.0025 to optimize the 30 600 parameters in DRRAiNN. To stabilize training, we clip the gradient if its norm exceeds 1, thereby preventing large parameter updates in steep regions of the loss surface. We use hydra to manage experiment configurations (Yadan, 2019).

To increase the size of the training data set and improve generalization, we apply data augmentation. The symmetry group of the square contains eight elements: the identity, rotations by 90, 180, and 270 degrees, and reflection in the x, y, and both diagonal axes. For each training sequence, we apply a uniformly sampled symmetry to the spatial variables in each time step. We ensure physical consistency by tapping into the runoff embeddings at the transformed station locations. The river discharge model's graph structure remains unchanged by this augmentation.

# 2.5 Benchmark model: European Flood Awareness System





To provide context for DRRAiNN's performance, we compare it to the European Flood Awareness System (EFAS), an established and operational distributed process-based model. We use publicly available EFAS reanalysis data, which eliminates the need to tune EFAS ourselves. This avoids potential biases that could arise from allocating unequal tuning effort to the benchmark model versus our own model. While DRRAiNN achieves higher performance than EFAS in many scenarios, our primary aim is to demonstrate the potential of distributed neural networks for river discharge estimation, rather than merely outperforming EFAS.

EFAS simulates runoff on an approximately  $1.5 \text{ km} \times 1.5 \text{ km}$  grid with a temporal resolution of 6 h. It receives as inputs static maps describing topography, river networks, soil, and vegetation, as well as meteorological forcings such as precipitation, temperature, and potential evaporation.

While EFAS serves as a useful benchmark, the comparison to DRRAiNN is not perfectly fair due to fundamental differences in the input and output variables. Both models receive gridded meteorological forcings, but DRRAiNN additionally receives discharge measurements during the tune-in period. In contrast, EFAS does not use discharge measurements as input but relies on them for offline model calibration. Furthermore, DRRAiNN produces discharge estimates only at gauging station locations, whereas EFAS generates discharge predictions across the entire spatial grid. EFAS also relies on additional input variables not used by DRRAiNN, such as soil type, vegetation, temperature, and potential evapotranspiration. While this makes EFAS a powerful tool, it also limits its applicability in regions lacking such detailed input data. Another difference lies in the precipitation data used: EFAS relies on EMO-1, a 6 mathrmh product interpolated from weather station data, whereas DRRAiNN uses RADOLAN, a radar-based dataset offering higher spatial and temporal resolution. As a result, a direct comparison between EFAS and DRRAiNN is not valid. Nonetheless, EFAS serves as a baseline to contextualize the expected performance range of DRRAiNN. We thus emphasize that our goal is not to directly compare performance but to provide a baseline that allows us to place the principled quality of DRRAiNN's performance with respect to alternative state-of-the-art forecasting approaches.

## 330 2.6 Evaluation







Besides visualizing hydrographs for selected gauging stations, we evaluate DRRAiNN using four standard metrics in hydrology: Kling-Gupta efficiency (KGE, (Gupta et al., 2009)), Nash-Sutcliffe efficiency (NSE, (Nash and Sutcliffe, 1970)), Pearson's correlation coefficient (PCC), and the mean absolute error (MAE). We report all four metrics because each highlights different aspects of model performance, and no single metric is free from limitations (Gupta et al., 2009). MAE is particularly intuitive, as it is expressed in the same unit as discharge and directly quantities the average deviation between predictions and observations. However, because it lacks normalization, stations with larger discharge magnitudes contribute disproportionately to the overall MAE. PCC quantities the strength of linear association between the observed and estimated discharges. While it captures shared variability, it is insensitive to systematic differences in scale or bias. To also capture the scale, the NSE was developed, which can be seen as a mean squared error that is weighted by the variance of the observed discharge. The NSE also does not account for bias, though, which is why the KGE was developed to jointly evaluate correlation, bias, and variability. When computing KGE and NSE values, we use station-wise means and variances calculated from the training data set, following the approach in Kratzert et al. (2019). For KGE, NSE, and PCC, higher values indicate better performance, with a maximum of 1 representing a perfect match. In contrast, lower values of MAE are better, with 0 indicating a perfect fit.

During open-loop inference, we evaluate metrics separately for each open-loop step, where the first step resembles closed-loop estimation. This allows us to assess how model performance degrades with increasing lead times. Although DRRAiNN was only trained on sequences that span 20 days, we evaluate it on 50 day sequences to investigate its ability to generalize beyond the training horizon. Additionally, we will plot the performance of the models against the mean discharge of the different stations to identify potential systematic dependencies between flow magnitude and model accuracy. In all cases, we exclude the initial 10 days tune-in period before calculating metrics and producing plots.

With knowledge discovery being the main motivation of this work, we also test DRRAiNN for physical plausibility. A physically implausible model might learn spurious relationships in the data. It could, for example, exploit the DEM to encode local biases that lead to gains or losses of water not driven by meteorological forcings. By retrospectively inferring catchment areas from observed dynamics, we assess whether the rainfall-runoff model successfully propagates water across the landscape. The procedure is as follows: After a forward pass, we compute saliency maps by taking the gradient of the final discharge estimate with respect to the precipitation inputs. These maps tell us to which extent the model's output depends on the precipitation in each grid cell and time step. We multiply this gradient by the precipitation itself to focus the analysis on cells in which precipitation occurred. To examine how the attributions change over time, we split the sequence into subsequences of 5 days over which we take the mean. We do this for each station separately and visualize the resulting attributions to identify which areas contribute most to discharge estimation at each station. To reduce noise, we repeat this process across all test sequences and average the resulting attribution maps.

We compare the resulting attributions with catchment areas delineated from elevation data using standard hydrological techniques, which are widely used in the field. To evaluate their agreement quantitatively, we employ the following measure when comparing DRRAiNN to the ablated models: For each station, the attributions are standardized to lie between 0 and

1 using min-max scaling. We then compute the Wasserstein distance between the attributions values inside the delineated catchment area and those outside it. A higher Wasserstein distance indicates better alignment between the attributions and the catchment areas delineated from elevation data. This quantitative measure complements the qualitative comparison, providing stronger evidence for our model's ability to propagate water across the landscape in a physically plausible way. Specifically, it indicates that the model has implicitly learned the topographic structure of flow direction – i.e., that water generally flows downhill – solely from observed discharge dynamics.

# 370 3 Results




To evaluate DRRAiNN, we first present hydrographs and compare performance with EFAS to contextualize DRRAiNN's results. We furthermore show that DRRAiNN can retrospectively infer catchment-like structures, thus demonstrating how full differentiability supports physical interpretability.

## 3.1 Hydrographs

EFAS produces hydrographs that match both the shape and magnitude of observed discharge, rendering it a strong contestant (Fig. 4). As EFAS produces gridded outputs, it is necessary to extract outputs from EFAS grid cells that correspond to the station locations in order to make meaningful comparisons.

DRRAiNN also produces plausible hydrographs that closely match the observed discharges. This includes both low flows (Fig. 4a) and high flows (Fig. 4d). No systematic difference in performance is observed across flow regimes. Since DRRAiNN operates autoregressively – using its own discharge estimates as input in the next time step – error can accumulate over time, leading to gradual decline in accuracy. Nonetheless, it is notable that the model is in general able to hit peaks even after almost 50 days, despite being trained only on 20 day sequences.

#### 3.2 Performance

Overall, DRRAiNN outperforms EFAS in all considered metrics (Fig. 5). Since EFAS does not incorporate discharge values during inference, we report its mean performance over lead times as constant. As described above, DRRAiNN's autoregressive nature causes errors to accumulate over time, leading to a gradual decline in performance at longer lead times.

The KGE plot (Fig. 5a) indicates that DRRAiNN is able to maintain strong performance over time. Averaged over the seeds, starting with a KGE of about 0.71, our model's estimations stay above those of EFAS during the entire estimation horizon of 50 days, despite having been trained only on 20 day sequences. In contrast, the NSE plot (Fig. 5b) shows gradual decline in performance over time with a decrease from 0.72 to 0.62 over the estimation horizon. Regardless, even after 50 days, all seeds show higher NSE values than EFAS. The PCC plot (Fig. 5c) shows a strong linear relationship between observed and estimated discharges, with an average value of about 0.9 at the start. DRRAiNN captures this relationship better than EFAS over the entire estimation horizon. Note that the linear correlation is also part of KGE and NSE. As the MAE allows direct

**Figure 4.** Hydrographs showing observed discharge, EFAS simulations, and predictions from one of five DRRAiNN model instances for lead times of up to 50 days. The four panels show the stations with the lowest (a) and highest (d) mean discharge, as well as the stations where EFAS (b) and DRRAiNN (c) achieve the best KGE performance on average on the validation set. For each station, we selected the sequence from the test set with the highest discharge variance, as variance likely serves as a proxy for prediction difficulty.

**Figure 5.** Performances of the best three out of five DRRAiNN model instances, compared to EFAS across different metrics and lead times up to 50 days. Results are averaged across all stations. Each line style corresponds to a distinct DRRAiNN instance.

interpretation, its plot (Fig. 5d) shows that EFAS is off by about  $6.5 \text{ m}^3 \text{ s}^{-1}$  on average, while DRRAiNN with  $3.9 \text{ m}^3 \text{ s}^{-1}$  on average on the first day produces a considerable smaller error. After about 25 days, EFAS yields a lower MAE on average.

All metrics reveal differences in performance across the model instances trained with different random seeds. However, the relative ranking of model instances varies depending on the specific metric and lead time. Some seeds perform better during the initial days, while others are better with greater lead times: For example, in the KGE plot (Fig. 5a), the ranking changes after about 42 days. The difference between instances are due to random weight initialization and the order of batches only. These stochastic factors may lead some instances to start the training with a larger bias towards capturing short-term, while others start with a larger bias towards capturing long-term relationships in the data.

The plots in Fig. 6 show that some stations consistently yield more accurate discharge estimates than others. This observation holds across all evaluation metrics. Which stations are harder to estimate, however, is different across the metrics, reflecting the distinct sensitivities each metric has, as discussed previously. Interestingly, both the different DRRAiNN instance and EFAS show partial agreement on which stations are more difficult to model. For example, the KGE values in Fig. 6a show that Altensteig and Stein are consistently easier to estimate, while Oppenweiler, Bad Imnau, and Murr are among the most challenging. The reasons for this discrepancy – such as differences in catchment size, land cover, or upstream complexity – could be analyzed in future work.

The regression lines indicate whether model performance correlates with average discharge levels across stations. We performed linear regression; the regression lines appear exponential due to the logarithmic scaling of the x-axis. All metrics,
except MAE, show that both models tend to perform better at stations with higher discharges. This effect is more pronounced
in EFAS, while our model exhibits a more balanced behavior. The differences between KGE and NSE patterns (Fig. 6a and b)
show that the models have different biases for the different stations, since KGE accounts for both bias and variability, while
NSE only captures variance. Both DRRAiNN and EFAS produce significantly larger MAEs with increased mean discharge
(Fig. 6d). This is expected since MAE does not account for the stations' mean discharges or their variability in discharge,
unlike the other metrics.

## 3.3 Catchment area inference






We observe that DRRAiNN implicitly infers physically plausible catchment areas, as shown in Fig. 7. Lighter areas indicate regions with higher importance of precipitation for estimating discharge at the corresponding station. These attribution patterns spatially overlap with the catchment areas delineated from elevation alone (depicted in red). The first column shows attributions averaged over the whole 20 day sequences. The remaining columns visualize attributions for subsequences of 5 days length to illustrate temporal changes in spatial influence.. There is a tendency of the area of influence to increases in size the further we look into the past. This suggests that DRRAiNN propagates encoded water quantities along the landscape in a manner that aligns, at least to some extent, with physical flow processes.

In the case of Pforzheim, DRRAiNN assigns low importance to an area in the lower right part, despite its inclusion in the delineated catchment area. This discrepancy could be related to known underground flows near Pforzheim, as reported in Ufrecht (2002). In the absence of subsurface flows, water would be expected to pass through Pforzheim; however, due

**Figure 6.** Performances of DRRAiNN and EFAS at a 1 day lead time across different metrics and stations. The x-axis shows the logarithmic mean discharge at each station. Blue vertical lines depict the standard deviation across DRRAiNN seeds. Dashed lines represent linear regressions between the log-mean discharge and corresponding metric.

**Figure 7.** Attribution maps of precipitation for discharge estimation at selected stations and time intervals, averaged over all test set sequences. Brighter colors indicate grid cells where precipitation has a stronger influence on the estimated discharge at the corresponding station. For comparison, traditional catchment areas delineated from elevation data are outlined in red. This juxtaposition highlights the agreement between data-driven attributions and physically derived catchment boundaries. The attribution method is described in detail in Subsect. 2.6 of the main text.

to the presence of underground flow paths, it instead moves towards the southeast, entering the Neckar River network via an alternative route. Our results suggests that DRRAiNN may have detected these unobservable underground flows from precipitation and discharge dynamics. However, this hypothesis arguably needs more investigation in the future.

Note that these results primarily serve as a proof of principle: We present results from the seed producing the clearest attributions; others yielded qualitatively worse results. However, it is important to keep in mind that DRRAiNN is trained on daily discharge measurements. Learning sharp catchment delineations would require the training data set to contain sequences in which it rained within the area, but not outside of it, over the extent of a 24 h period. As precipitation is very dynamic on this time scale, the chances for this are relatively low. In the future, we expect sharper results if we go from daily to hourly discharge data.

# 3.4 Ablations







To assess both the physical plausibility and contributions of specific architectural components, we conducted a series of ablations on DRRAiNN (Appendix A). First, we showed that DRRAiNN can exploit the DEM as a positional encoding by training, validating, and testing it on a rotated DEM. However, it did result in slightly worse performance and less physically plausible behavior (Appendix A1). Next, we evaluated the model's inductive bias in distinguishing between spatially extended and local processes (Appendix A2). Last, we removed the hypernetworks to examine their impact (Appendix A3). Both ablations led to performance degradation across most metrics and lead times. However, the differences were not always significant. Importantly, neither ablated model was able to produce physically realistic catchment areas, as demonstrated both qualitatively and quantitatively.

## 4 Discussion

We introduce DRRAiNN, a fully differentiable, fully distributed neural network architecture for estimating river discharge from past discharge, gridded elevation maps, and gridded precipitation and solar radiation. DRRAiNN demonstrates better performance than EFAS on lead times of up to 50 days. This indicates that DRRAiNN can produce valid estimations far into the future despite it being trained on sequences of only 20 days, including a warm-up period of 10 days.

Our analysis reveals that the difficulty of discharge estimation varies across gauging stations. Interestingly, both DRRAiNN and EFAS consistently struggle with the same stations, suggesting that the difficulty is intrinsic to the stations and their associated data rather than specific to the model architecture. Several factors likely contribute to this variability. For example, stations affected by unobserved variables such as complex subsurface topography, land cover heterogeneity, or anthropogenic factors (e.g., dam operations) may be inherently harder to model. Furthermore, spatial variations in the quality of input data could contribute to discrepancies in performance. Future investigations using attribution techniques could offer deeper insights into these station-specific variations and guide the development of architectural modifications or regularization to address these challenges effectively.

Our ablation studies show the benefits of distinguishing between spatially extended and local processes, and of incorporating hypernetworks. The reduced performance and failure of the ablated models to produce realistic catchment areas suggests that these components encode crucial hydrological processes, such as water movement across complex topographies. This suggests that incorporating appropriate inductive biases can both improve model interpretability and reduce the risk of learning spurious correlations.

Interestingly, the model instance that produces the most physically plausible attribution maps is not the one with the best predictive performance. This points to a trade-off between optimizing for predictive accuracy and encouraging physically realistic model behavior. This suggests that conventional performance metrics, while effective at evaluating predictive accuracy, may not fully reflect whether the model adheres to underlying physical principles.

Increasing the amount of training data generally enhances performance in ML. Currently, DRRAiNN is not designed for scalability, as its application is expected to require retraining in each specific context. A natural step toward improving adaptability would be training DRRAiNN on hourly discharge data. This could improve performance and attribution quality, potentially enabling the model to trace the origins of individual discharge peaks. Since traditional PBMs rely on a wider range of input variables, feeding them as additional inputs could also lead to performance improvements in DRRAiNN. This includes land cover, parent material, soil texture, vegetation, temperature, and potential evapotranspiration among others. Interpretability methods can then be used to perform a sensitivity analysis, revealing which input variables are important when and, due to our model being fully distributed, where. These methods may also provide insights into the model's internal representations, potentially uncovering links to real-world hydrological variables.

Several strategies can be employed to investigate DRRAiNN's spatial generalization capabilities. One approach is to leave out individual stations within a river network during training to evaluate generalization within hydrologically connected regions. A more demanding test of generalization would involve training and testing on different river networks. By testing it on catchments that are not part of the training data, we can systematically assess its ability to generalize to unseen regions. Ultimately, we aim to apply DRRAiNN to diverse catchments across Germany, Europe, or globally. Due to DRRAiNN's data-driven nature, discharge measurements will always be needed for training. However, recent advances in remote sensing may enable the application of DRRAiNN to ungauged river networks (Gigi et al., 2019).

#### 5 Conclusions




In this paper, we introduced DRRAiNN, a fully distributed neural network architecture that estimates river discharge from precipitation, solar radiation, elevation maps, and past discharge measurements from gauging stations. Despite being trained on sparse target data – namely daily discharge observations from 17 stations over ten years – DRRAiNN outperforms the operational benchmark model EFAS in terms of KGE and NSE across various lead times. Beyond its predictive accuracy, DRRAiNN provides physically interpretable attributions, enabling the identification of precipitation sources contributing to discharge at specific stations. Our analyses highlight the importance of incorporating hydrologically meaningful constraints, or inductive biases. These biases not only enhance interpretability but also help the model align more closely with physical

principles, as evidenced by its ability to delineate realistic catchment areas. With its predictive performance, interpretability, and physical consistency, DRRAiNN represents a promising step forward in the application of neural networks to distributed hydrological modeling.

Code and data availability. The preprocessed data sets can be found at Scholz et al. (2025a). The code can be found at Scholz et al. (2025b).

## **Appendix A: Ablations**

## A1 Rotated elevation map

We aim to assess whether DRRAiNN utilizes the elevation map in a physically plausible way – specifically, to propagate water downhill across the landscape. An alternative would be that DRRAiNN leverages the elevation map primarily as a positional encoding, allowing it to orient itself within the landscape and learning location-specific biases. In practice, both mechanisms are likely at play to some degree.

To examine this, we train, validate, and test DRRAiNN using the same elevation map as before, but rotated by 180 degrees. This setup preserves the statistics of the elevation map, ensuring a fair comparison.

For most metrics and lead times, DRRAiNN performs better when trained and tested on the original elevation map compared to the rotated one (Fig. A1). Nonetheless, its continued superior performance relative to EFAS – even with the rotated DEM – supports the hypothesis that DRRAiNN leverages elevation as a positional encoding. Remarkably, this still enables it to reconstruct plausible catchment areas to some extent (Fig. A2). However, our quantitative analysis (Fig. A3) shows that catchment areas are more accurately reconstructed when DRRAiNN is executed on the original DEM. This suggests that our original model's use of the elevation map goes beyond mere positional encoding, incorporating hydrologically meaningful information.

#### 510 A2 All LSTM



A key inductive bias in DRRAiNN is the explicit separation between spatially extended processes and local processes. Lateral water movement across the landscape is a spatially extended process primarily driven by elevation. Evapotranspiration, on the other hand, is a local process that is largely influenced by solar radiation. We encode this distinction into DRRAiNN by assigning these processes to different components of the ConvNeXt block: the DWConv is parameterized by a CNN that receives elevation as input, while PWConv1 and PWConv2 are parameterized by an MLP that receives solar radiation. In this ablation, we discard this bias by feeding the elevation and solar radiation – together with precipitation – directly into the PWLSTM. Consequently, the relativity bias, realized by subtracting the elevation of the center cell from the elevations of all other cells within each receptive field of the hypernetwork, is also removed.

We observe a significant performance drop for all metrics except MAE (Fig. A4). In addition, the inferred catchment areas appear less plausible compared to those produced by DRRAiNN (Fig. A5), a finding that is supported quantitatively (Fig. A6).

**Figure A1.** Performances of the best three out of five DRRAiNN model instances and DRRAiNN model instances on a rotated elevation map, compared to EFAS across different metrics and lead times up to 50 days. Results are averaged across all stations. Each line style corresponds to a distinct DRRAiNN instance.

**Figure A2.** Attribution maps of precipitation for discharge estimation at selected stations and time intervals, averaged over all test set sequences with a rotated elevation map. Brighter colors indicate grid cells where precipitation has a stronger influence on the estimated discharge at the corresponding station. For comparison, traditional catchment areas delineated from elevation data are outlined in red.

**Figure A3.** Wasserstein distances between normalized attributions inside and outside the catchment areas delineated from the digital elevation model. A higher distance indicates better agreement between inferred and delineated catchment areas, suggesting more physically realistic model behavior. Standard deviations are computed across the different gauging stations.

These results demonstrate that explicitly distinguishing between spatially extended and local processes benefits DRRAiNN in terms of both predictive accuracy and physical plausibility.

# A3 No hypernetworks



Here, we train DRRAiNN without hypernetworks to assess their contribution. To stay close to the original architecture, we preserve inductive bias that distinguishes between the spatially extended process of water propagation and the local process of evapotranspiration. Specifically, the elevation map is concatenated with the hidden state, passed through a position-wise linear layer, and then fed into the DWConv. This step is necessary because DWConv requires the input and output channels to be of equal size. As a result, the relativity bias, realized by subtracting the elevation of the center cell from the elevations of all other cells within each receptive field of the hypernetwork, is also removed. For solar radiation, we concatenate it with the hidden state and feed the result directly into PWConv1.

Removing the hypernetworks from DRRAiNN results in decreased performance for KGE and NSE (Fig. A7a and A7b). For PCC and MAE, we do not observe a systematic difference (Fig. A7c and A7d). The ablated model produces less plausible attributions maps compared to DRRAiNN (Fig. A8), a finding that is supported quantitatively (Fig. A9).

## **Appendix B: Alternative hyperparameters**

In this appendix, we report the performance of DRRAiNN under alternative hyperparameters settings. In the default configuration, the LSTM in the rainfall-runoff model has a hidden size of 4, and the GRU in the discharge model has a hidden size of

**Figure A4.** Performances of the best three out of five DRRAiNN model instances and DRRAiNN model instances where all forcings are fed into the PWLSTM, compared to EFAS across different metrics and lead times up to 50 days. Results are averaged across all stations. Each line style corresponds to a distinct DRRAiNN instance.

**Figure A5.** Attribution maps of precipitation for discharge estimation at selected stations and time intervals, averaged over all test set sequences when all forcings are fed into the PWLSTM. Brighter colors indicate grid cells where precipitation has a stronger influence on the estimated discharge at the corresponding station. For comparison, traditional catchment areas delineated from elevation data are outlined in red.

**Figure A6.** Wasserstein distances between normalized attributions inside and outside the catchment areas delineated from the digital elevation model. A higher distance indicates better agreement between inferred and delineated catchment areas, suggesting more physically realistic model behavior. Standard deviations are computed across the different gauging stations.

8. Here, we examine DRRAiNN's performance using both smaller and larger hidden sizes. Additionally, we assess the impact of replacing the GRUs in the discharge model with LSTMs.

## B1 Rainfall-runoff model with hidden size 2

Figure B1 shows that reducing the hidden size of the rainfall-runoff model from 4 to 2 still yields a competitive model. On average, it performs slightly worse during the initial days. However, due to the variance in performance across different seeds, additional experiments are required to draw a more definitive conclusion.

#### **B2** Rainfall-runoff model with hidden size 6

Figure B2 shows that increasing the hidden size of the rainfall-runoff model from 4 to 6 slightly decreases performance on the NSE and PCC metrics, while KGE remains largely unaffected. Since no significant improvement is observed, we argue that the smaller model should be preferred, following Occam's razor.

## **B3** Discharge model with hidden size 4

Figure B3 shows that reducing the hidden size of the discharge model from 8 to 4 significantly reduces performance across all metrics and lead times.

**Figure A7.** Performances of the best three out of five original DRRAiNN model instances and DRRAiNN model instances without hypernetworks, compared to EFAS across different metrics and lead times up to 50 days. Results are averaged across all stations. Each line style corresponds to a distinct DRRAiNN instance.

**Figure A8.** Attribution maps of precipitation for discharge estimation at selected stations and time intervals, averaged over all test set sequences without hypernetworks. Brighter colors indicate grid cells where precipitation has a stronger influence on the estimated discharge at the corresponding station. For comparison, traditional catchment areas delineated from elevation data are outlined in red.

**Figure A9.** Wasserstein distances between normalized attributions inside and outside the catchment areas delineated from the digital elevation model. A higher distance indicates better agreement between inferred and delineated catchment areas, suggesting more physically realistic model behavior. Standard deviations are computed across the different gauging stations.

## **B4** Discharge model with hidden size 16


Figure B4 shows that increasing the hidden size of the discharge model from 8 to 16 leads to mixed results. While KGE appears to deteriorate, NSE and PCC show slight improvements, particularly at longer lead times. Since no significant improvement can be observed, we argue that opting for the smaller model align better with Occam's razor.

## **B5** Discharge model with LSTM

Figure B5 shows that replacing the GRUs in the discharge model with LSTMs significantly reduces performance across all metrics and almost all lead times. This suggests that model complexity should reflect the complexity of the underlying dynamics: river flow tends to follow simpler dynamics than surface and subsurface flow, which we model with an LSTM. Moreover, water typically resides in channels for shorter periods compared to its residence time below ground. This may explain the superior performance of GRUs in the discharge model, though further investigation is warranted.

Author contributions. All authors contributed to the conceptualization of the paper. FS, MT, and MB designed the model architecture. FS developed the code and performed the experiments. FS prepared the manuscript with contributions from all co-authors.

Competing interests. The authors declare that they have no conflict of interest.

**Figure B1.** Performances of the best three out of five original DRRAiNN model instances and DRRAiNN model instances with a hidden size of 2 in the rainfall-runoff model, compared to EFAS across different metrics and lead times up to 50 days. Results are averaged across all stations. Each line style corresponds to a distinct DRRAiNN instance.

**Figure B2.** Performances of the best three out of five original DRRAiNN model instances and DRRAiNN model instances with a hidden size of 6 in the rainfall-runoff model, compared to EFAS across different metrics and lead times up to 50 days. Results are averaged across all stations. Each line style corresponds to a distinct DRRAiNN instance.

**Figure B3.** Performances of the best three out of five original DRRAiNN model instances and DRRAiNN model instances with a hidden size of 4 in the discharge model, compared to EFAS across different metrics and lead times up to 50 days. Results are averaged across all stations. Each line style corresponds to a distinct DRRAiNN instance.

**Figure B4.** Performances of the best three out of five original DRRAiNN model instances and DRRAiNN model instances with a hidden size of 16 in the discharge model, compared to EFAS across different metrics and lead times up to 50 days. Results are averaged across all stations. Each line style corresponds to a distinct DRRAiNN instance.

**Figure B5.** Performances of the best three out of five original DRRAiNN model instances and DRRAiNN model instances with LSTMs instead of GRUs in the discharge model, compared to EFAS across different metrics and lead times up to 50 days. Results are averaged across all stations. Each line style corresponds to a distinct DRRAiNN instance.

Acknowledgements. We thank the reviewers for their very constructive criticism, feedback, and suggestions. This work received funding from the Deutsche Forschungsgemeinschaft (DFG, German Research Foundation) under Germany's Excellence Strategy – EXC number 2064/1 – Project number 390727645 as well as from the Cyber Valley in Tübingen, CyVy-RF-2020-15. Additional support came from the Open Access Publishing Fund of the University of Tübingen. The authors thank the International Max Planck Research School for Intelligent Systems (IMPRS-IS) for supporting Fedor Scholz and Manuel Traub. ChatGPT was partially used to improve the writing style of the manuscript.

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
