# Peer review of "Fully differentiable, fully distributed Rainfall-Runoff Modeling"

_EGUsphere, 2024_

## Author Response (AR1)

**Reply to: RC1: 'Comment on egusphere-2024-4119', Shijie Jiang, 23 Mar 2025**

This is a thoughtful and well-executed study. The authors propose DRRAiNN, a fully differentiable and fully distributed neural architecture for rainfall-runoff modeling. The model design is novel and ambitious. In general, the model is interesting and many components are well motivated, while several parts of the paper would benefit from clearer framing, stronger justification, and more focused discussion. Below are my detailed comments and suggestions.

Thank you so much for this exhaustive feedback. The referee raised several points that will substantially improve the quality of our manuscript. In the following, we will reply to each of these points in detail.

1. In the Introduction and Related Work, the authors use a large portion (to line 67) to review process-based and data-driven hydrological models. However, much of this content is already well-established and could be condensed. More importantly, the link between the challenges described in the background and the specific research goal of DRRAiNN is not clearly established. If the main goal is to improve the interpretability of ML models or to incorporate physical constraints into neural architectures, then the extended discussion of PBM limitations seems unnecessarily long. The core research motivation (around line 74) is somewhat buried between the general background and the introduction of differentiable modeling.

If the intended contribution is to implement a distributed hydrological model using NNs, then the literature review does not sufficiently acknowledge recent progress in this direction. In the Related Work section, the authors list several NN-based rainfall-runoff models, but the discussion is somewhat narrow. For instance, the paper states that "not many" fully distributed data-driven models exist (without citation), but several relevant models with explicit routing modules have been proposed in the last two years (e.g., https://doi.org/10.1029/2023WR036170, https://doi.org/10.1029/2023WR035337, https://doi.org/10.1016/j.jhydrol.2024.132165). These are not acknowledged or compared.

I suggest shortening the background section, clearly identifying the research gap, and referencing recent distributed or physically guided models to more accurately position DRRAiNN in the current landscape. Importantly, to clarify the contribution, it would be helpful to include a concise statement explaining why a fully differentiable, fully distributed rainfall-runoff model is needed and what specific challenge it addresses, as motivated by prior work.

We agree that Introduction and Related Work should be shortened and that our motivation and contributions should be made more clearly. Our main goal is to create fully differentiable, fully distributed rainfall-runoff model. This means that the model learns from data and does not rely on assumptions regarding catchment area delineation or drainage directions. Thank you for providing those references! They present interesting approaches that need to be respected when discussing differentiable, distributed hydrological models, which is why we will incorporate them into our Related Work section. However, these models have stronger assumptions, which restricts them with regards to possible flow directions, and are therefore not fully distributed as we understand it: https://doi.org/10.1029/2023WR036170: Here, water always flows towards the steepest descent, which turns the grid into a large tree. Unobservable flows in different directions cannot be captured. - https://doi.org/10.1029/2023WR035337 and https://doi.org/10.1016/j.jhydrol.2024.132165: These approaches are based on NNs that receive basin-averaged attributes and forcings. Combining the different basins in a tree-like structure is what we would call a semi-distributed approach. Unobservable flows between basins in different directions cannot be captured. The attribution maps where not the main motivation of our work, however, they are a good example of an interpretability method and show that our model can indeed infer the outline of catchment areas from observed dynamics. We think showing this is important, as a distributed model is only valuable if it makes adequate (i.e. physically plausible) use of the distributedness, which is not at all guaranteed when working with neural networks. In a future version of our manuscript, we will shorten the background on process-based and data-driven models, incorporate the provided references, and clearly state our motivation and contributions.

- 2. I like the general idea behind the model using separate neural networks to handle spatial and temporal dynamics makes a lot of sense. I do have a few suggestions:
- I) One thing I found a bit confusing is the use of the term "runoff." Even though the authors explain it is not the physical runoff, it still might be misleading. The variable comes from the LSTM's hidden state and goes through several layers, including a CNN that does not consider flow direction or water accumulation. So it is more like a learned feature than an actual variable. Maybe calling it something like "runoff embedding" or "runoff representation" would make things clearer.

Thank you for pointing that out. We agree that the variable should be referred to as "runoff embedding", as this underlines its self-organized nature.

II) The model uses solar radiation to model local ET. I get the idea, but I am not fully convinced it is enough. Radiation tends to be spatially smooth, especially at the catchment scale. I would expect that including vegetation-related variables (like LAI, NDVI, or GPP) could help better capture spatial variability in ET.

We agree that vegetation-related variables are potentially very useful to better

model ET. However, our goal for now was to create a minimalistic model that can learn from the little target data that we have. We hope that it inspires more research in the future that makes use of such variables.

III) A few model choices could use more explanation. For example, why use hidden size 4 for the LSTM and 8 for the GRUs? These seem small for a model working across space and time.

Most hyperparameters were chosen based on trial and error, i.e. by looking at the validation performance. We started with small hidden sizes as this reduces parameters count and therefore reduces the risk of overfitting. However, we understand that some of these choices seem arbitrary. We will perform further analyses, which might be reported in the appendix of a future version of our manuscript.

IV) The link between the rainfall-runoff module and the discharge model (although I think alternative terms might help avoid confusion with traditional concepts) seems functionally effective but conceptually weak. Right now, it is unclear what information the embedding actually carries for downstream discharge estimation, and whether it supports realistic routing behavior. Some clarification on what this embedding is supposed to represent in hydrological terms would be helpful.

This is connected to the point raised above (renaming "runoff" to "runoff embedding"). We agree that it is unclear what information exactly the embedding carries, since it is fully self-organized by the neural network. We know, however, that it contains information about runoff, as otherwise the model would not have been able to implicitly infer the catchment area outlines. Please also have a look at the supplementary material of this comment, which underlines this ability further. However, without further examination, it is difficult to make more precise statements about its meaning. In future work, it would therefore be very interesting to "open the box" and make use of interpretability methods to get a better understanding of what the neural network is doing here.

V) Since the model emphasizes interpretability, it might be useful to consider whether the internal states could reflect more structured hydrological components. For example, separating fast and slow flow signals, or introducing latent variables that relate to soil moisture or baseflow. I understand the authors may not follow a process-based philosophy, but some explanation of why those processes were not treated as explicit model components, while lateral propagation was, would be helpful.

While we know that components like fast and slow flow, soil moisture, and baseflow exist, their separation is not always clear. For example, there might be states "in between" fast and slow flow. In order to avoid problems that might arise from such assumptions, we want our model to automatically learn

about these components and to treat them in a self-organized manner. Lateral propagation on the other hand is a critical ingredient for a distributed model. In general, we think that the main task of the modeler is to incorporate those biases that best restrict the solution space without introducing wrong assumptions. We aim to clarify this in a future version of our manuscript.

3. I understand the idea behind using symmetry-based data augmentation for generalization in purely statistical terms, but I am not sure if it makes hydrological sense. Rotating or flipping the DEM and precipitation might result in flow directions that are not physically meaningful, especially since the river network and station layout stay fixed. Some clarification or discussion around this would be useful.

Data augmentation is always performed across all variables consistently, including the station coordinates, which is why flow directions remain meaningful. We will state this more clearly in a future version of our manuscript.

4. In model evaluation, one thing that could make the analysis stronger is to also report metrics for specific types of hydrological conditions — for instance, during rising limbs, low flows, or flood events. This would help clarify whether the model is merely capturing average behavior or actually learning the dynamics that matter most.

We will consider metrics that explicitly measure our model's performance for specific types of hydrological conditions, like FHV and FLV. We want to point out, however, that the main motivation of our work is to create a model that extends our understanding of hydrological processes in general. While we hope that it can also serve as a starting point for more practical applications like flood forecasting, it is currently not our focus.

5. The current attribution analysis focuses entirely on where the rainfall matters spatially. However, hydrological response is also highly dependent on timing (e.g., time to peak). It might be worth considering how the model distributes attention over past precipitation steps, or whether it systematically over- or under-reacts to delayed signals. Even a simple attention plot or error histogram over lag times could be insightful.

Thank you so much for that suggestion. We performed a more detailed analysis that takes time into account when creating the attribution maps. Please have a look at the supplementary material. It essentially shows that cells that are farther away impact the station's discharge later in time. We think that this further underlines the physically plausibility our model and will certainly add these results to our manuscript.

6. The comparison of hydrographs in Fig. 4 only provides a qualitative perspective. I would suggest including some quantitative metrics to support the

statements made in Sect. 4.1. For example, it is mentioned that the large peak on day 80 in Lauffen and Rockenau is underestimated by both models - is this meant to suggest that the error comes from the input data (e.g., underestimation in the precipitation forcing)? If so, it would be helpful to state this explicitly. Otherwise, it could still be related to limitations in the model architecture or the autoregressive setup. Also, only one model instance is shown in the figure. I wonder how consistent the five trained seeds are - do they show similar hydrographs, especially in the later part of the prediction window, or is there high variability? Some indication of uncertainty or model spread would help.

We will consider further metrics in a future version of our manuscript. The fact that the peak is underestimated by both models could be related to errors or inaccuracy in the input data, but also to shortcomings that both models have in common (e.g. spatial resolution). We will state this more clearly and suggest to examine it further in future work. We decided to only plot a single hydrograph as we wanted to find and present the best model based on the validation data set. In general though, we agree that it would be useful to create ensembles of predictions to get an uncertainty estimate.

- 7. There are a couple of claims in Section 4.2 that could benefit from clarification or more substantive support.
- I) The authors mention that different seeds lead to different behaviors, with some instances performing better on short lead times and others on long lead times. It is not clear how weight initialization alone would systematically bias a model toward short- or long-term prediction. From a modeling perspective, are there specific components (e.g., LSTM, GRU) that are more sensitive to initialization in this regard?

The sensitivity to weight initialization is a shortcoming of our model that should be examined and improved in the future. Where in the model this sensitivity is situated is not clear at this point, but most likely related to the components that handle the temporal aspects of the data, namely LSTM and GRU. We will make sure to mention this in our manuscript. Additionally, by now we have some new results with a much smaller spread in performance. We might use these new results in an updated version of the paper.

II) The explanation that some stations are more difficult to model due to unobservable underground flows or pipes feels vague and speculative (lines 401-407). Since the authors already emphasize that DRRAiNN is distributed and physically interpretable, it would be more convincing to check whether these "hard-to-predict" stations differ in observable properties in their controlling catchments, such as elevation range, forest cover, drainage density, geology (e.g., using map overlay analysis with the HydroATLAS dataset).

Thank you for this very nice idea. We removed any assumptions from the

sentence and mention the idea for future research work.

- 8. I find the idea of reconstructing catchment areas from saliency maps very interesting. However, I have a few concerns about the framing and the strength of the conclusions in Section 4.3:
- I) The attribution map shown is only for a single seed, with the justification that all seeds were temporally validated. However, attribution is often highly sensitive to parameter noise, especially for gradient-based methods. It would strengthen the argument to show whether the attributions are consistent across seeds, or to provide a measure of saliency variance.

As we show in Appendix A3, the attribution maps are indeed not quite consistent across seeds. Our paper emphasizes from the beginning that the task to build a DRRAiNN-like system with very restricted target value data (daily discharge) is indeed very challenging. As a result, we did not succeed in generating an algorithm that is fully guaranteed to converge. Nonetheless, we used the best seed in terms of validation error to generate the attribution maps.

II) While it is interesting that a known discrepancy exists, there is no real evidence that DRRAiNN "discovered" the underground flow. A more conservative interpretation might be that the model failed to align with the delineated catchment, and this could be due to unmodeled processes. If the authors want to keep this discussion, it would help to at least show whether the model consistently de-emphasizes that region across multiple sequences.

We agree that the potential discovery of underground flows could use more evidence here. As discussed above, in future work the hidden states could be examined in more detail for a better understanding of the model's inner workings, which could underline such a statement. However, it should be pointed out that the attribution maps are averaged over the whole validation data set as described in the Methods section. The behavior of the model is therefore quite consistent here.

III) In Figure 7, I noticed that most of the examples shown are for stations in smaller headwater catchments. It would be helpful to also evaluate downstream stations, where we would expect the model to aggregate signals over a broader upstream area. If the attributions remain very local in those cases, it might suggest that the model is not truly learning large-scale accumulation, but rather reacting to recent local precipitation.

The supplementary material shows attribution maps for Lauffen, a downstream station. Here, the attributions spread much farther, taking upstream stations into account as desired.

IV) More generally, it is unclear whether the attribution reflects actual learned

flow dynamics, or just highlights locations where recent rainfall occurred. A simple test might be to check whether attribution strength correlates with rainfall intensity rather than hydrologically relevant pathways. This would help clarify whether the model is truly learning how water moves through the network, or just where it rains.

Since the attributions are computed by multiplying the gradient with the rainfall itself, there will certainly be a correlation between those quantities. However, we think that the supplementary material underlines that the model actually learned flow dynamics, at least to a certain extent.

- 9. The discussion is rich and touches on many aspects of the model's behavior and potential. However, it currently reads more like a collection of loosely connected observations and future directions (e.g., abrupt shifts from ablation to data choices to flood forecasting), rather than a focused and structured analysis. Some points are speculative without clear support, while others deserve more detailed treatment.
- I) The authors mention that DRRAiNN is "not designed for scalability," but it remains unclear what limits its scalability: is it due to computational costs, architectural complexity (e.g., the combined grid and graph operations), or something else? It would help to provide a clearer picture of the computational resources and time required for training and inference.

Our comment on scalability aimed at different spatial and temporal resolutions. To which extent our model is scalable in this sense is an open question for future work. The limiting technical factor will most likely be the available memory on a graphics card. However, memory usage depends on multiple hyperparameters, which might be adapted for the specific application. Training took less than 8 hours on a single A100 graphics card. In a future version of our manuscript we will clarify what we mean with scalability and add information about the required computational resources.

II) The discussion includes many potential future directions (e.g., hourly discharge, new inputs, removal of the warm-up phase, ....). While these are all interesting, I would recommend narrowing the focus to 1-2 directions that are most promising or directly tied to the current model's limitations.

We agree that a more condensed outlook would be beneficial and will respect this in a future version of our manuscript.

III) The observation that the best attribution map does not correspond to the best predictive performance is interesting. However this claim may be based on a small number of seeds, and attribution can be sensitive to initialization. Could this divergence be due to noise or model variance?

Yes, this divergence could very well be due to noise and should be examined further in the future.

**Reply to: CC1: 'Comment on egusphere-2024-4119', Benedikt Heudorfer**

Great work. It adds to the increasing body of literature of spatially distributed ML models in hydrology. You might be interested in recent research by Martin Gauch as well.

Thank you very much for your feedback! We will certainly have a look into

Alyway, regarding the manuscript I have one minor comment and one non-binding suggestion.

First, in line 162 you state that you "derive static topography features from the digital elevation model". It does not get entirely clear what kind of static features you use. I "classic" entity-aware LSTM setup, usually some derived values are used. I interpret it that you simply use the DEM grid-specific elevation as such? Please specify what exactly you do here.

Thank you for pointing that out. Yes, we use the DEM grid-specific elevation as such. However, within the model, we perform a small computation: In each 7x7 receptive field of the CNN, we subtract the elevation of the center cell from all other cells, which turns the elevation into local differences of elevation. We describe this in the model section (lines 242-244). This is all we wanted to hint at with "derive static topography features". We will simply remove it from there.

Second, benchmarking against a process-based model is fine, but process-based and conceptional hydrological models have long been outmatched by DL models in performance, including by multiple studies you cite, and can not be called benchmark anymore. I don't question the obvious good performance of your model (figure 5), but to really showcase what it can do, it should be additionally compared to a regular entity-aware LSTM. Implementing this in a similar, multihorizon setup should be straight-forward. This suggestion is non-binding, but would greatly improve the significance of this work.

Thank you for the suggestion. The focus of our study is to show that a distributed neural network is in general able to learn catchment outlines from observations and propagate water accordingly. Good performance was not our main goal. Due to time restrictions, we therefore did not prioritize the implementation of a lumped model, which represents a very different modeling approach. However, we are looking into options.

**Reply to: RC2: 'Comment on egusphere-2024-4119', Peter Nelemans**

**General comments**

The authors present DRRAiNN, a ML rainfall runoff model capable of predicting streamflow at multiple locations. The model architecture is interesting, as it is designed to incorporate certain inductive biases. The model shows good performance, and a notable highlight of the study is the authors' approach to identifying which grid cells influence the simulated discharge at specific locations. Overall, the study is well-executed, but at times, the manuscript appears to lack important details, which could pose challenges for readers attempting to build upon this work.

Thank you very much for your very extensive and detailed comments! We think addressing the raised issues will significantly improve our manuscript. In the following, we will reply each point in more detail.

**Specific comments**

One of my main issues with the manuscript is its characterization of DRRAiNN as a fully distributed model, even though discharge is estimated only at specific stations. While the input is fully distributed and the rainfall runoff predicts embedded runoff in a distributed manner, no attempt at interpretation of these embeddings was done, so I assume they may not be interpretable. Consequently, if the only physically meaningful output is generated at discrete locations, the model should in my opinion be classified as semi-distributed. I suggest that the authors either provide additional justification for considering DRRAiNN fully distributed or reclassify it as a semi-distributed model.

Thank you for this suggestion. It seems to us that what "fully distributed" exactly means is a matter of definition. It is certainly true that our model does not produce a runoff estimation in each grid cell. However, the supplementary material shows that our model propagates quantities along the landscape, indicating that the individual grid cells actually contain information about runoff. How this information could be directly translated into runoff is not at all obvious, since we don't have measurements in each grid cell. So, in a sense, we think that our model is already "as distributed as can be," at least when it comes to modeling the real world. We are considering adding the term "river discharge" to the title of our manuscript to emphasize the fact that the estimated quantities are situated within the rivers.

Another one of my concerns with the manuscript is the absence of any reporting or discussion on the computational efficiency. Although it does not have to

be a large part of this study, it remains relevant. This is even more so when considering scaling up DRRAiNN to larger areas, increasing temporal or spatial resolution, or applying the model in flood forecasting scenarios where many weather forecast ensembles must be processed in short time periods.

This is important, so we will add this information to our manuscript. While the training is of course time consuming (but manageable with only 8 hours on an A100), the time it takes for the model to estimate a day of discharge is in the order of milliseconds.

**1. Introduction**

The introduction is generally well-written. I suggest that the authors consider emphasizing the "dirty little secret of hydrology": the common assumption that there is no leakage into or out of a simulated catchment through the subsurface. This issue could potentially be addressed by DRRAiNN and represents one of the most interesting and significant contributions of the study.

Thank you, we will emphasize this aspect more.

The authors present the model as fully differentiable. Here, "differentiable" does not refer to the ability to perform gradient-based optimization, but rather to a specific approach of combining physics with ML, as defined by Shen et al. (2023, see https://doi.org/10.1038/s43017-023-00450-9). The authors do not elaborate on the distinction between a model that is simply "differentiable" and one that is "fully differentiable", nor do they explain why DRRAiNN qualifies as the latter. While I am not suggesting that it is not, I suggest that the authors provide a clearer explanation of these terms and justify labelling DRRAiNN as fully differentiable, especially given that this feature is highlighted in the manuscript's title.

With stating that we are presenting a "fully differentiable" model we meant to emphasize "end-to-end" differentiability, meaning that gradients can flow seamlessly through the entire system, enabling optimization of all its parts. We will clarify this in the future version of our manuscript.

Notwithstanding the two aforementioned (non-binding) suggestions, I find the introduction to be somewhat lengthy. Although there is in my opinion not a specific paragraph that must be removed in its entirety, I do suggest condensing most, if not all, paragraphs to some extent.

As the other referee and we ourselves agree with this, we will condense the Introduction and will avoid giving too much background.

Lines 30-31: A lumped model implies spatial averaging at the basin scale but does not necessarily imply temporal averaging. This is also clarified later in the

manuscript (lines 116–118)

Thank you, we removed time from that sentence.

Lines 31-32, "Therefore, the outline .... be feasible.": The outline of the basin must always be available for PBM-based catchment modelling, regardless whether one employs a lumped, semi-distributed, or fully distributed approach.

We suggest to differentiate between the approximate and exact outline. Indeed, we also needed know the approximate catchment area of the Neckar as a whole to be able to place our grid. However, with our approach, we do not need to know the exact outline of each sub-basin, as this can be inferred by the model itself. We will adjust our manuscript accordingly.

Lines 49-50, "...for which no detailed land...": I suggest replacing "no" with "few".

Lines 59-60, "[ANNs] high complexity also often leads to neural networks not respecting physical laws despite very good performance": I would argue the complexity of ANNs is not the reason they have difficulty to respect physical laws. Rather, their data-driven approach is.

Lines 69-70, "[A] considerable amount of runoff is situated below the ground and therefore not observable": I suggest clarifying a bit: "[A] considerable amount of runoff is often situated below the ground and therefore not directly observable."

Line 70, "It is not yet possible to see through the ground": I suggest rephrasing to: "It is not yet possible to directly measure subsurface flow."

Line 78, "A combination of the above-mentioned approaches could leverage the advantages of both worlds": I suggest to clarify that you are referencing PBMs and ML here.

Line 96, "Another one": I suggest to clarify: "Another inductive bias"

Thank you for these very detailed suggestions. We incorporated all of them into our manuscript.

**2. Related work**

This section is well-written but does not sufficiently address the relevant literature. Given that the paper focuses on ML-based hydrological modelling, I suggest expanding this section to include additional studies that explore models with architectures other than LSTM and CNN-LSTM. Without this, readers might mistakenly get the impression that the field has only explored these particular architectures.

Specifically, I suggest discussing studies on the use of neural ODEs (e.g., Höge et al., 2022, https://doi.org/10.5194/hess-26-5085-2022), as these models focus strongly on knowledge discovery and system understanding. Furthermore, I suggest including studies that propose physics-informed, fully distributed ML models. E.g.: Sun et al. (2022, https://doi.org/10.5194/hess-26-5163-2022) and Chen et al. (2022, https://doi.org/10.1145/3534678.3539115). Otherwise, this section might unintentionally imply that the pursuit of physics-informed ML models is a recent development.

Thank you for these suggestions. Unfortunately, we were not fully aware of this additional literature, but think it is important to always consider alternative approaches. We will integrate them accordingly into our manuscript and point out the differences to our approach.

Line 121, "[F]ully distributed models operate on a grid without any spatial aggregation": In some cases, such as in this study, the forcings are still aggregated, despite the inputs being used in a fully distributed manner.

We are not certain whether we understand this comment correctly. Does it point to that fact that runoff is collected, and thereby aggregated, at the station locations? An alternative interpretation is the fact that we are coarsening our grid before feeding it into the model. In both cases, we will remove the "spatial aggregation" from this sentence.

Line 124: I would specifically mention that the model presented by Xiang and Demir (2022) is a GNN, especially given the use of a GNN in this study. Readers might be familiar with GNNs, but not with the work by Xiang and Demir (2022). That being said, I do not think there is a need to go into detail. A simple addition would suffice. E.g.: "The GNN-based model presented by Xiang and Demir (2022) indeed..."

Perhaps this is a matter of personal preference, but I suggest placing the authors' names outside the parentheses when referring to them directly. E.g. for line 130: "In contrast, Oddo et al. (2024) used a..."

Thank you for these suggestions. We incorporated them into our manuscript.

**3. Method**

I suggest starting this chapter with an introduction to the model and moving what is currently section 3.3 to the beginning. I have several reasons for this. First, the DRRAiNN model is the central focus of the paper, so I would begin the chapter with its introduction. The study site is less relevant to most readers and, in my opinion, would be better placed later in the chapter.

Secondly, reordering the sections in this way would improve the overall flow.

The current description of the DRRAINN model is, as it should be, general and independent of the study area or data sources. After introducing the model, the study site and data sources can be presented, as they are more specific. This creates a natural lead into section 3.4, the experimental setup. The chapter would thus progress smoothly from the general workings of the model to the specific details of the experiment, rather than jumping between the two.

Lastly, introducing the data after the model would provide better context. For example, in lines 170-171, the data used to construct the adjacency matrix is mentioned. However, this is also the first time the adjacency matrix is introduced, and its function only becomes fully clear in section 3.3.

Thank you for this nice suggestion! We definitely agree that a reordering would improve the flow and will implement this change in our revision.

As it stands, the exact role of the different kinds of data in the model is, in my opinion, sometimes difficult to ascertain. For instance, while it can be inferred that solar radiation and precipitation together form the dynamic meteorological forcings F, this is not explicitly stated. Similarly, the static maps S, which, as far as I can tell, only include the DEM, suffer from the same issue.

We agree and will make better use of the introduced notation to improve intelligibility.

I also suggest to add a sentence on the study area's climate and some of the characteristics of the river itself (mean, max, min discharge).

We will add some information about the catchment's characteristics to our manuscript.

Lines 152-153: Temperature is typically included in hydrological models for two main reasons: to model snow (and glacier) processes, and to model evaporation. Since snow is a rare occurrence in the study area, as far as I can tell (which highlights the importance of including climate details for the catchment), the lack of improvement when including temperature in the input data may be more related to the specific study area than to the model's internal mechanics.

Additionally, solar radiation is strongly correlated with temperature, so for evaporation modelling, it might be sufficient to include either solar radiation or temperature, as both could potentially inform the model's prediction of evaporation. While I have no issue with the approach or the sentence itself, I suggest adding some further explanation.

We assume that our model did not improve when adding temperature is due to the correlation between temperature and radiation, as mentioned. However, we can see that it is not obvious why we chose radiation over temperature in the first place. This design choice was driven by the intention to model evapotranspiration as well. Nevertheless, we will perform another experiment where we replace radiation with temperature.

Figure 1: Please include a colour bar for the DEM. For the x- and y-axes, I recommend using latitude and longitude, as the kilometre scale on both axes serves no function given its arbitrary origin, other than acting as a scale bar (which should therefore be included when switching to a lat-long axis).

Additionally, the river network is shown in great detail with a consistent width, which makes it difficult to discern which segment a hydrological station belongs to, as the red dots indicating the station locations sometimes overlap multiple streams. To improve clarity, it might help to omit small side streams and/or adjust the width of the river segments according to, for example, the Strahler order.

Furthermore, the grey arrows are not included in the legend. While I assume they are meant to indicate the hierarchy between the stations in relation to the flow of the river, I believe they make the figure more confusing rather than clearer. These arrows may be redundant and could be removed if the river network were illustrated more clearly, as I suggested earlier.

Lastly, the figure is not referenced in the text. While this might be personal preference, I suggest to include a reference to the figure, for example, in line 141.

Thank you, these are some very nice suggestions. We will invest more time in improving the figure and implement your recommendations.

Line 157: I suggest mentioning the temporal resolution as well.

Lines 160-161: For additional clarity, I suggest to specifically mention that the mean and standard deviation are computed after having added 1 and taking the logarithm.

Line 161: I suggest using "zeros" instead of "0s". Additionally, the fact that 0 is the mean is a result of the standardization process, not the logarithmic transformation. Therefore, I suggest to refer to 0 as the "standardized mean" rather than the "log-standardized mean".

Lines 163 and 168: The authors mention using rasterio to transform data to the RADOLAN CRS twice. For conciseness, I suggest stating this once at the end for both datasets.

Line 167: I suggest mentioning the spatial and temporal resolution.

Thank you, we will implement all of these suggestions in our manuscript.

Lines 175-176: If the discharge measurement station locations are corrected manually, it seems unnecessary to snap them to the river network afterward. I would expect the corrected locations to already be on the river network.

The station locations were corrected manually by using multiple sources of information, partially from different institutions and maps. Snapping them to the river network actually changed their coordinates, but only to a very small extent.

Line 188: I suggest removing "future", since historical forcings are used.

Line 206: I suggest clarifying that the process described between lines 200-206 is repeated per timestep in an auto-regressive manner.

Line 234: The authors could consider adding an equation for the PWConv as well, next to Equation 2.

Thank you again, we will implement these suggestions in our manuscript.

Lines 253-255: Why did the authors choose to implement two kernels, and effectively employ a GNN with two types of nodes? Why not opt for a single node type (the stations), using river segment length and elevation difference as edge features between station nodes? A GRU could still be implemented, before, after, or during the message-passing iterations. I am wondering what motivated the authors to adopt this specific (and rather complex) model architecture? Was a simpler setup unable to achieve the desired results, or was there a conscious design decision behind this choice? I suggest elaborating on this part of the model architecture.

The recurrent graph neural network is based on previous work from our lab, a model called DISTANA (Karlbauer, Matthias, et al. "A distributed neural network architecture for robust non-linear spatio-temporal prediction." arXiv preprint arXiv:1912.11141 (2019)). We will add a reference to this work in our manuscript. The idea is to have transition kernels whose main job is to aggregate and model the exchange of information between nodes, while the station kernels maintain information at the nodes themselves. In general, it is perfectly reasonable to try other kinds of graph neural networks here. We decided to use one that we already established before.

Line 261: Initially, two types of kernels are introduced: station and segment kernels. However, line 261 refers to a "transition kernel". Based on Figure 2, I assume this is the same as the segment kernel. If so, I suggest consistently using the term "segment kernel" to avoid confusion. If my assumption is incorrect, then I suggest clarifying what is meant by "transition kernel".

Thank you, this was relict from renaming the components during the writing process..

Lines 261-262, "Each kernel first concatenates its static, dynamic, and lateral inputs...": Based on Figure 2, it appears that the segment kernel only receives static inputs (adjacency matrix, elevation difference) and lateral inputs (from neighbouring stations). While these lateral inputs are dynamic in nature, the phrasing suggests a distinction between "dynamic" and "lateral" inputs. If "dynamic inputs" refer specifically to discharge and embedded runoff, then only the station kernels receive them. I suggest clarifying this distinction or rephrasing to avoid confusion.

Here, we described the architecture in a more general way than we actually used it in our experiments. We will clarify this.

Figure 2: I really like this figure, especially how the rainfall-runoff component is visualized and the connection between the two models is illustrated. However, I suggest adding more detail on the types and dimensions of the data streams, as these are not always clear from the text. For example, between the linear block and the StationGRU, you could add the label "Embedded runoff" and indicate its dimensions (e.g., Hx,y,c). I also recommend clarifying that the discharge data in the bottom left is observed discharge, while the discharge in the bottom right is simulated discharge. It would also be helpful to include dimensions for all input data, such as solar radiation, precipitation, altitude difference, river segment lengths, and the adjacency matrix.

Thank you, this will certainly make the figure more comprehensible.

Line 264, "Afterward, the tensor is split into dynamic and lateral outputs.": I suggest clarifying that the dynamic outputs refer to the estimated discharges, while the lateral outputs are embeddings used to update the SegmentGRU again, as indicated by the two outgoing arrows from the StationGRU in Figure 2.

Line 291: I suggest that the authors specify that they refer to a NVIDIA A100.

Lines 292-297: I suggest the authors provide further explanation regarding their choice of hyperparameters. For instance, were other optimizers considered besides Ranger? Did they explore the use of learning rate schedulers? What criteria were used to determine the sizes of the hidden states? Additionally, why was the SiLU activation function selected?

It is no problem if these decisions were made without any detailed exploration of alternatives, as the focus may have been on demonstrating the proof of concept rather than exhaustive model optimization. However, some more information about these choices (or the lack thereof) could help future researchers in determining what works well and what has not been tried yet (and thus where there

is potentially still room for improvement).

Most of the hyperparamters were found experimentally. Indeed, we tried different optimizers, played around with learning rate scheduling, and tried different activation functions. Currently, we are still investigating the effect of different hidden sizes. In general, if two options produced similar results, we settled for the simpler one. Learning rate scheduling, e.g., did not improve the results and was therefore removed.

Lines 326-327: I am unsure whether the inclusion of four different metrics adds value to the study, particularly if they are conceptually similar. For example, KGE consists of three components, one of which is PCC, and is essentially the same as NSE but without a certain bias. I suggest either replacing NSE and PCC with other, more distinct metrics or omitting them entirely.

Thank you, we agree and will happily remove the NSE and PCC from the manuscript.

**4. Results**

The results are described and illustrated in detail. However, aside from the four (relatively similar) metrics, the performance of DRRAiNN is not compared to that of EFAS in any other way. Additionally, the inferred contributing area per discharge station, as simulated by the DRRAiNN model, is compared to the catchment area delineated from the DEM. As the authors correctly point out in the introduction, this DEM-based delineation can be problematic, particularly due to its disregard for underground flows.

Nevertheless, a mismatch between the DEM-delineated catchment and the contributing areas from DRRAiNN is sometimes framed as an error by DRRAiNN (e.g., lines 420, 425). As the authors correctly note (lines 426-429), this mismatch does not necessarily indicate an error, but could reflect an unobserved subsurface flow path. I suggest presenting both the DEM-delineated catchment and the contributing area from DRRAiNN as feasible options, rather than assessing one against the other.

We agree, this could be phrased better. When we say that the results are not perfect, we mean the fact that in some cases, a square is visible which corresponds to the receptive field of the CNN. We will describe this in more detail and consider both as reasonable options.

If the authors wish to include a more objective assessment of the physical plausibility of the DRRAiNN model, they could train the model on EFAS-simulated (or any other PBM model) discharge instead of observed discharge. Since the contributing area is known a priori in this case (the DEM-delineated catchment), training DRRAiNN on EFAS would ideally result in a perfect alignment between

the contributing area according to DRRAiNN and the DEM-derived catchment. While such experiments may be outside the scope of the current manuscript, they could be explored in a follow-up study.

Furthermore, I had hoped to see in this chapter some analysis of the internal system states, especially given their limited number (either 4 or 8) and they're location in GRUs or LSTMs. It's conceivable that certain hidden states respond to specific inputs, which could make them at least partially physically interpretable. While I don't expect this to be included in the current study, it's a valuable direction the authors could consider for future research.

Both of these are very interesting endeavors that we considered but didn't find the time yet to pursue. We will make sure to mention them in the future work.

Lines 369-371: "We likely chose ... considered stations." I suggest leaving this line out, as in my opinion it is redundant.

Agreed.

Figure 5: Nice figure! I suggest including the unit  $(m^3/s)$  in the MAE plot. Also, if I understand correctly, the DRRAiNN performance is based on the three best-performing seeds. I would suggest explicitly mentioning this in the caption.

The large difference in performance across different seeds is interesting. With better learning rate scheduling, this issue could potentially be mitigated or at least reduced. Bentivoglio et al. (2023, see https://doi.org/10.5194/hess-27-4227-2023) employed a curriculum learning strategy, similar to the TBPTT used here, and found that gradually lowering the learning rate every few epochs improved performance.

Thank you, we tried learning rate scheduling in the past but didn't not find a significant difference in performance (or spread thereof). However, currently we running some promising experiments that do not seem to suffer from this. We might be able to put these new results into an updated version of our manuscript.

Line 406: In lines 442-444, the authors (correctly) list some factors that could contribute to the difficulty in predicting streamflow at certain stations beside unobserved subsurface flows, but these are not listed here. I suggest listing these also (or only) here.

Referee comment RC1 suggested that an analysis of these factors would be interesting, with which we agree. As this is out of the scope of this study, though, we will most likely keep them in the future work section and remove them from the results section.

Figure 6: I find the information in this figure very interesting, but the figure

itself could be improved. While this is subjective, the figure might look more visually appealing if the performance at each station were represented as markers rather than by a line connecting the performance per station. The range of DRRAiNN performance across different seeds could be indicated with whiskers. Additionally, the mean discharge per station could be plotted as a dashed line behind the performance markers for better visibility of the markers.

Yes, in retrospect it makes no sense to connect the stations. We will update the figure accordingly.

Figure 7: This is a very interesting figure and, in my opinion, showcases one of the most important contributions of this study. However, I believe the figure could be improved in several ways. First, the individual plots are quite small and difficult to read; I suggest making them larger. The DEM-based catchment area is barely visible. Additionally, displaying the same legend in all four plots is in my opinion unnecessary; I recommend having a single legend, placed outside the plots.

The station name is already in the title of each plot, so repeating it inside the plot is redundant and makes the plot harder to read. Instead, I suggest indicating the station of interest with a different marker.

Furthermore, though this is subjective, I believe the plots would benefit from being zoomed in on the station and its catchment area. There is no need to display the entire river network, as the goal of this figure is comparing the DEM-derived catchment and the attribution area from the DRRAiNN model. Therefore, it would be sufficient to zoom in per station and show the station, the DEM-derived catchment area, the attribution area, and possibly the river network within the catchment.

Finally, as an ambitious suggestion (and possibly challenging to implement in 2D), it would be fascinating to see the attribution overlaid with the DEM itself. This would provide insight into how the area identified by DRRAiNN as significant aligns with the DEM.

Again we agree, the plots can be improved substantially. Thank you for the concrete feedback, we will surely make use of it.

**5. Discussion**

I recommend expanding the discussion to address whether and how future work could overcome the limitation of estimating discharge only at observation stations. More importantly, I suggest to discuss whether the model could even become fully independent from discharge observations, which would expand its applicability to ungauged basins. These are two important drawbacks of the current model, but neither are addressed explicitly.

We discuss a little bit of spatial generalization, but agree that this could be extended. In some preliminary experiments, we simply treat individual stations from the training data set as "virtual" stations. This means that these stations are part of the graph, but the model never receives discharge as input or target at these stations. It seems that the network can predict discharge at these locations to some extent already. Another option would be to remove the graph neural network completely from the model. However, some form of target data will always be required. These must not necessarily come from gauging stations, though, as there are efforts to estimate discharge from satellite observations. We will make sure to discuss these aspects in our future work section.

Furthermore, I suggest reordering the discussion somewhat as to group related paragraphs together. E.g. lines 485 – 489 discuss including more input data, and line 494-499 discuss including more output data.

Additionally, the overall discussion is quite long, and I suggest trying to condense some paragraphs. Line 447-453 could maybe be moved to the chapter 4.

Lines 462-470: In my opinion, this should be part of section 3.2.

These suggestions are in line with Referee Comment RC1. In a future version of the manuscript, we will try to condense the discussion, especially the future work.

Line 489: A warm-up period of 10 days is quite an achievement! Most models, especially large and complex PBMs, often require warm-up periods of a year or longer. A model with such a short warm-up period is impressive, and I believe the authors should highlight this accomplishment more explicitly.

Thank you! We were not aware of that and are happy to highlight this in our discussion.

Lines 491-492: Assuming the authors are referring to remotely sensed soil moisture, this would be limited to the moisture in the upper soil layers, not the deeper, saturated layers. Therefore, I believe the initial 10-day warm-up period is still necessary, even with soil moisture as an input, since the model will also need time to adjust the deeper soil moisture. However, including compressed meteorological data might proof effective.

Again a valid point that we were not aware of. We might remove this point from the discussion in an effort to condense.

**6. Conclusion**

The conclusion is well written and concise.

Thank you!

Appendix A

This is overall a very nice addition to the study, and similarly well written.

Figure A1: To be consistent with Figure 5 and Figure 6, I suggest to plot the data from EFAS again in orange.

Figure A2: This is very interesting, and I believe plotting the entire region is now warranted, as the attributed cells are located far outside the DEM-derived catchment. However, I still suggest increasing the figure size, using a single legend for all four plots placed outside the plots, and differentiating the stations with distinct markers rather than displaying their names.

Lines 543-544: If I understand correctly, the entire ConvNeXt block has been removed from the architecture. I suggest explicitly stating this for clarity.

Figure A4 and Figure A7: same comment as for Figure A1.

Figure A5 and Figure A8: same comment as for Figure A2.

In the ablation studies described in A2 and A3, significant portions of the model are removed, which likely results in fewer model parameters and faster runtimes. I suggest discussing the model size and computational efficiency in this context as well.

We will certainly improve the quality of our figures, including the ones in the appendix. You are correct, the model has fewer parameters and different runtimes. We will look those up and report them in our revision.

Technical corrections

Line 290: Missing "table".

Figure 4: Missing "discharge" after "... and highest (d)".

Thank you!

**Reply to: CC2: 'Comment on egusphere-2024-4119', Tianfang Xu**

This is a well-executed, novel study demonstrating the potential of CNNs and GNNs for spatially distributed rainfall-runoff modeling.

However, we would like to clarify that our cited works (lines 131-135, Longyang et al., 2024; Xu et al., 2022; Tyson et al., 2023) were specifically designed for karst watersheds where subsurface connectivity is watershed-specific, often unknown, and largely independent of topography. The spatial attention mechanism in Longyang et al. (2024) was explicitly developed, in combination to ridge regression, to identify contributing areas and lateral flow paths that cross topographic boundaries, with results validated against hydrogeochemical tracers.

Additionally, these studies were watershed-specific implementations (as clearly stated in their objectives) and were not presented as generalizable to ungauged basins without additional training. Our ongoing work shows these methods perform well when trained on other watersheds, suggesting their transferability potential.

Given these differences in scope and methodology, we respectfully request either:

A more precise characterization of our work's context and contributions, or Removal of the citations if they do not align with the discussion of global aggregation approaches.

We appreciate the opportunity to provide this clarification and would be happy to discuss further if helpful.

Thank you very much for your comment and for pointing that out. We want to sincerely apologize for describing your approach in a way that may have come across as overly critical. We actually find your approach interesting and think it makes a lot of sense! Our intention was not to suggest that your model is incapable of detecting underground flows - this may well be something our models have in common. What we aimed to highlight is that your model does not have a direct incentive to propagate water along the grid cells. Since it takes a weighted mean over the whole catchment, the water can effectively "jump" from any location to the outlet. This is something a lot of ML-based full distributed models from the literature have in common. In contrast, our model is forced to propagate the information along the grid, more or less from one cell to another. As this differentiates our model from others, we wanted to emphasize this aspect. We will make sure to express this point more clearly in the revised version and will remove the last sentence about physical plausibility

and generalization. Are we overlooking anything in this regard? Please don't hesitate to let us know!

**List of relevant changes made in the manuscript**

- 1. Introduced a previously unseen test set, which results in lower performance for both EFAS and DRRAiNN, but more clearly highlights DRRAiNN's superior performance. Due to limited data availability, only one year was available for testing. To prevent large individual discharge peaks from causing abrupt jumps in the metric-over-time plots, we reduced the maximum estimation horizon to 50 days. This ensures that such peaks appear once per lead time, resulting in smoother and more interpretable curves. All model outputs that would have been included in a 100-day plot are also contained within the 50-day plot.
- 2. Improved clarity by rephrasing and refining the figures as suggested by the reviewers.
- 3. Introduction
  - Removed well-established background information.
  - Incorporated related work, which we condensed and expanded, including references suggested by the reviewers.
  - Clarified the motivation for a model like ours.
  - Better emphasized our contributions.
  - Precisely defined what fully distributed means in our context.
- 4. Reorganized the Methods section into the following structure: (1) Model, (2) Data, (3) Study site.
- 5. Added more information on study site's characteristics.
- 6. Added information on compute time.
- 7. Included new results based on the test set.
- 8. Added attribution maps over time and for a larger catchment.
- 9. Explicitly communicated the limited stability of the attribution maps.
- 10. Condensed the Future Work section significantly.
- 11. Added an appendix with results for alternative hyperparameter settings.

---

## Referee Report (RR1)

Note from the reviewer: The current manuscript is a revised version of original submission by the authors, dated 7th of April, 2025, which I reviewed on 8th of April, 2025. Therefore, I will keep my comments brief, as the authors have already addressed the majority of my earlier concerns. I focus here mainly on a few remaining points.

**General comments**

The authors present DRRAiNN, a ML rainfall runoff model capable of predicting streamflow at multiple locations. The model architecture is interesting, as it is designed to incorporate certain inductive biases. The model shows good performance, and a notable highlight of the study is the authors' approach to identifying which grid cells influence the simulated discharge at specific locations. Furthermore, the manuscript is well written, with a clear and fluent structure. It is concise, focused, and now addresses several important aspects that were missing in the earlier version.

**Specific comments**

One of my main issues with the previous version of the manuscript was the authors classification of DRRAiNN as a fully distributed model. I appreciate that the authors now clarify why they consider their model fully distributed. Although I personally still disagree, and would classify the model as semi-distributed, I can accept their reasoning.

My other main issue with the previous version of the manuscript was the lack of reporting on the computational efficiency of DRRAiNN. I appreciate the authors adding this information. Although it is discussed only briefly, computational efficiency is not the focus of the study, and the current reporting on it is in my opinion sufficient.

**1. Introduction**

The introduction is well written and of high quality. Section 1 "Introduction" and section 2 "Related work" of the previous version of the manuscript have been merged into a single section in the revised version. Taking into account this merger, I appreciate the authors condensing the introduction.

The authors have substantially expanded their discussion of related work, offering a more comprehensive and focused overview of recent machine learning developments in hydrological modelling. I appreciate that rather than merely increasing the number of studies cited, they focus on those most relevant to this study. I also highly appreciate the way the related works are discussed: the authors succeed in both tracing the broader development of ML applications in hydrology over recent years and situating their own contribution within this narrative.

Line 58: At small-scale, a lysimeter can be used to directly measure overall evaporation (evapotranspiration).

Lines 107-109: I appreciate the authors clarifying why the model classifies as fully differentiable, which was one of my issues with the previous version of the manuscript.

At the end of the introduction, I suggest adding a brief sentence noting that the model is compared to the EFAS model for the Neckar River. This can be very concise and will help set the reader's expectations

**2. Methods**

This section is well written, and I appreciate the authors' efforts in rearranging the subsections. The overall flow has improved significantly, making the presentation more natural and providing the reader with additional context where needed.

The authors have added a brief introduction to this section, which was missing in the previous version of the manuscript. While the idea of including an introduction is useful, the current text does not effectively set up the content of Section 2. It reads more like a condensed abstract, lacking a clear overview of the topics covered in this section and referencing material that is actually discussed in the following section, which may confuse the reader.

Lines 121-122, "We present ... distributed manner.": I suggest clarifying here that this section, specifically subsection 2.1, provides a detailed examination of the model's architecture and the rationale behind key design choices, since a general introduction to the model was already given in the previous section. Currently, lines 121-122 simply repeat a condensed version of lines 103-107 and do not clearly indicate what the reader should expect from this section. Additionally, it would be helpful to note that the input and output data of DRRAiNN will be discussed in this section, as these aspects are closely related to the model's architecture.

Lines 122-123, "We evaluate ... design choices.": This is not discussed in this section, but rather in section 3. I suggest to remove this sentence.

Lines 123-124, "We demonstrate ... Awareness System.": The actual demonstration of model performance is part of section 3 "Results". In this section the study area, experimental setup, benchmark model, and evaluation metrics are introduced, and this can be mentioned here in the introduction of the section. They are essentially all components required to assess performance, but the results themselves are presented in the following section.

Line 124-126, "DRRAiNN achieves ... modeled dynamics.": This is a summary of your results and should not be mentioned in the methodology.

Lines 131-132, "... a grid that spans the whole catchment area of the river network.": Some readers might mistakenly assume the static maps and meteorological forcings span *only* the DEM-delineated catchment area, which is common practice. However, one of the main motivations for developing DRRAiNN is that the effective catchment may extend beyond the DEM-delineated area, and DRRAiNN therefore also takes a larger domain as input. I suggest clarifying that the input data covers a larger domain that includes the DEM-delineated catchment area, but also extends beyond it.

Lines 141-142: I appreciate the authors clear description of the nature of the input, internal states, and output of DRRAiNN.

Lines 148-149, "Despite being ... self-organizing nature.": I appreciate this important clarification, as it resolves a point that was unclear to me in the original version of the manuscript.

Figure 1: I appreciate the improvements made to this figure, which now provides a clear and solid understanding of the inner workings of the model, especially in combination with the text. My only remaining concern is the two-sided arrow beneath the box labelled "StationGRU," whose meaning is not yet clear.

The description of the rainfall–runoff model (subsection 2.1.1) is somewhat difficult to follow, though this is understandable given the model's complexity, and I appreciate the authors' effort to explain it as clearly as possible. In contrast, the description of the discharge model (subsection 2.1.2) is clear, concise, and highly effective

Lines 195-196: I find this very interesting; I was already wondering why one would only aggregate embedded runoff at the stations and not at every upstream river grid cell.

I appreciate the thorough and detailed description of the data used in the study, provided in subsection 2.2. The data preprocessing, as well as the temporal and spatial resolution, are clearly reported, addressing one of my concerns with the previous version of the manuscript.

Line 257: Although I appreciate the authors' description of the temperature data, the extreme values convey limited information about local climate conditions. I suggest replacing these with, for example, the mean summer and winter temperatures.

Figure 3 has improved considerably, and the issues I noted with this figure in the previous version of the manuscript have been adequately addressed.

Line 259: I highly appreciate the authors' description of the discharge characteristics.

Lines 281-284: I appreciate the authors pointing this out, and I agree that it is entirely reasonable to leave the evaluation of DRRAiNN under forecast-based conditions for future work.

Line 292: Changing the batch size during training, to counteract the increased memory use by the simultaneously changing truncation length is a very elegant solution! I appreciate the authors highlighting this aspect of their work.

Line 293: I appreciate the authors mentioning the computational time for a forward pass, as this was not mentioned in the previous version of the manuscript.

Line 300-304: Reading between the lines, I understand that a CNN trained only on the original data may perform worse when presented with a rotated or reflected version of the data. To address this and improve the CNN's generalization, I understand the authors train it on various symmetries of the data. If this interpretation is correct, I suggest adding a brief sentence explaining the issue of data symmetries in CNNs. Additionally, although it is somewhat mentioned in line 304, it may be helpful to more explicitly state that GNNs do not suffer from this problem.

Lines 315-326: I appreciate the authors' detailed explanation of the differences between EFAS and DRRAiNN, not only in terms of the models themselves but also their inputs, outputs, resolutions, and use cases. I also value their acknowledgment that the comparison of DRRAiNNperformance with EFAS is not entirely valid. However, as the authors clearly state, this is not their intention; their goal is not to outperform EFAS, but to provide a baseline comparison. In my opinion, their reasoning is entirely fair and justifies the comparison made.

Subsection 2.6: I appreciate the authors providing their reasoning for including each of these metrics, as I previously suggested reducing the number of metrics. With this explanation, I understand their rationale for using four different metrics.

Lines 356-357: The NSE does include the bias, but "in normalized form scaled by the standard deviation" of the target variable (Gupta et al., 2009, see https://doi.org/10.1016/j.jhydrol.2009.08.003). That was one of the main motivations to

develop the KGE, which incorporates the correlation, bias, and variability independent from each other.

**3. Results**

The reporting of results is clear and thorough, and the figures are well-designed. Especially Figure 6 and Figure 7 have much improved compared to the previous version of the manuscript. I also appreciate the thoughtful interpretation of the results and model performance, including the explanation for the varying performance of the DRRAiNN model across different seeds and lead times.

My only significant issue with this section, which I also mentioned in my previous review, is the sometimes seemingly conflicting assessment of the attribution maps. I agree that, to a large extent, the attribution maps from DRRAiNN are expected to overlap with the DEM-delineated catchment areas and could thus be used as an indicator of physical plausability. However, as the authors themselves point out, subsurface flow can, in some cases, transcend DEM-delineated catchment boundaries.

Therefore, I suggest that the authors exercise caution when classifying a DRRAiNN model as better solely because its attribution maps show more overlap with traditional catchment delineations than another DRRAiNN model (lines 461-464). To a certain extent, overlap between the DRRAiNN attribution map and the DEM-derived catchment area is expected and is indeed an indicator of physical plausibility. However, beyond a certain point, increased overlap cannot be assumed to correlate with higher physical plausibility.

In case an attribution map is wildly different from the DEM-delineated catchment, it should indeed be assessed lower than one with more overlap. However, once the differences between two DRRAiNN models are small, caution is needed. As the authors note, DRRAiNN may be capable of detecting unobserved subsurface flow paths, though, as they also emphasize, further research is required to confirm this.

I understand the delicate balance between these potentially contradictory considerations, but I urge the authors to avoid rigorously stating that more overlap automatically indicates a better model, especially since they acknowledge that the DEM-delineated catchment map may not capture all flow paths. When the authors state that one DRRAiNN model instance has more overlap than another and should therefore be regarded as having more physical plausibility (lines 461-464), I suspect they are referring to cases where the lack of overlap from the latter model cannot be explained by subsurface flow alone. If that is the case, I suggest clarifying this for the reader.

If the lower overlap could also be due to undetected subsurface flow paths, I suggest the authors refrain from jumping to conclusions and remain open to the possibility that DRRAiNN may have identified such paths (as, for example, noted in lines 426–427).

Lines 404-405: The quality of the observational data at these locations may influence the performance of both models. The authors are in a better position to assess this. If they consider this a potential reason for the reduced model performance at these locations, I suggest mentioning it here

Lines 410-411: The NSE does include bias and variability, see Gupta et al. (2009, https://doi.org/10.1016/j.jhydrol.2009.08.003).

Line 415: I assume the authors intended to indicate that darker areas correspond to regions of higher importance.

I appreciate the extensive series of ablations conducted by the authors.

**4. Discussion**

The discussion is well written, providing a clear summary of the work as well as several interesting suggestions for future research. I appreciate the authors' efforts in condensing the discussion.

**5. Conclusion**

The conclusion is well written and concise, and it clearly highlights the added value of the study.

**Technical corrections**

Line 39: A period is missing after "biases".

---

## Author Response (AR3)

**Reply to: Referee #1: Jiang, Shijie**

We would like to thank the reviewer again for their valuable feedback and address each point separately in the following.

I went through the revised manuscript and the author's response carefully. Most of the big concerns I raised in round one have been addressed, and I appreciate the effort. A few minor items remain that in my view still need attention before the paper can be accepted.

1. The Introduction refers to "integrated gradients" (l117), while the Methods actually describe input  $\times$  gradient saliency. Please remove for clarity.

Even though usage of integrated gradients is generally possible with DRRAiNN, we used saliency maps in our approach. We replaced the term and citation accordingly.

2. 1376, "no systematic difference" == ¿ But no event-based metric is presented, just by visual judgment is not convincing.

We agree and removed the sentence.

3. You consistently highlight "best three out of five seeds." This reads optimistic. I'd prefer to see averages and spread over all five seeds in text, and then keep the three-seed curves for visualization.

We appreciate this concern. Our seed selection is performed exclusively on validation data (not test data), with test evaluation only after selection to prevent cherry-picking. We believe this approach is methodologically sound because it reflects practical deployment where practitioners select the best validation performance, and we are transparent about evaluating five seeds and selecting three. We acknowledge that averaging all seeds would provide different statistical information. On the other hand, we do explicitly discuss initialization sensitivity as a limitation. Our current approach demonstrates the model's potential when properly initialized rather than averaging across suboptimal initial weight configurations.

Our analyses have shown (and we have confirmed this over the last year over and over again) that suboptimal configurations are rare but unfortunately persist most likely due to the low target data volume (i.e., actual discharge measurements). Reporting 5 out of 5 distorts the performance due to the rare outliers. This is why we think that it is better to stick to the best 3 out of 5 choice using the validation set (not the test set, which we only use to assess and report the performance - that is, the performance that is reported in the paper). We hope that the reviewer agrees that this is the better choice in the end. Thank you

for your consideration.

4. l239, please specify the value of "predefined threshold"

Thank you for pointing this out, we explicitly added the threshold of 1km.

5. The phrase "fully distributed" in the title/Abstract could confuse readers, since the model outputs only at gauges. Maybe qualify this up front in the Abstract to avoid overselling.

We added to the abstract that DRRAiNN estimates river discharge at gauging stations.

**Reply to: Referee #2: Nelemans, Peter**

We would like to thank the reviewer for their repeated extensive feedback. All points raised are valid and were addressed, which we think again improved the quality of our manuscript.

Note from the reviewer: The current manuscript is a revised version of original submission by the authors, dated 7th of April, 2025, which I reviewed on 8th of April, 2025. Therefore, I will keep my comments brief, as the authors have already addressed the majority of my earlier concerns. I focus here mainly on a few remaining points.

**General comments**

The authors present DRRAiNN, a ML rainfall runoff model capable of predicting streamflow at multiple locations. The model architecture is interesting, as it is designed to incorporate certain inductive biases. The model shows good performance, and a notable highlight of the study is the authors' approach to identifying which grid cells influence the simulated discharge at specific locations. Furthermore, the manuscript is well written, with a clear and fluent structure. It is concise, focused, and now addresses several important aspects that were missing in the earlier version.

Thank you!

**Specific comments**

One of my main issues with the previous version of the manuscript was the authors classification of DRRAiNN as a fully distributed model. I appreciate that the authors now clarify why they consider their model fully distributed. Although I personally still disagree, and would classify the model as semi-distributed, I can accept their reasoning.

My other main issue with the previous version of the manuscript was the lack of reporting on the computational efficiency of DRRAiNN. I appreciate the authors adding this information. Although it is discussed only briefly, computational efficiency is not the focus of the study, and the current reporting on it is in my opinion sufficient.

**1. Introduction**

The introduction is well written and of high quality. Section 1 "Introduction" and section 2 "Related work" of the previous version of the manuscript have been merged into a single section in the revised version. Taking into account this merger, I appreciate the authors condensing the introduction.

The authors have substantially expanded their discussion of related work, offering a more comprehensive and focused overview of recent machine learning developments in hydrological modelling. I appreciate that rather than merely increasing the number of studies cited, they focus on those most relevant to this study. I also highly appreciate the way the related works are discussed: the authors succeed in both tracing the broader development of ML applications in hydrology over recent years and situating their own contribution within this narrative.

Thank you so much!

Line 58: At small-scale, a lysimeter can be used to directly measure overall evaporation (evapotranspiration).

Thank you, we now state that it is difficult to measure spatially distributed evapotranspiration.

Lines 107-109: I appreciate the authors clarifying why the model classifies as fully differentiable, which was one of my issues with the previous version of the manuscript.

At the end of the introduction, I suggest adding a brief sentence noting that the model is compared to the EFAS model for the Neckar River. This can be very concise and will help set the reader's expectations

We agree and added a corresponding sentence.

**2. Methods**

This section is well written, and I appreciate the authors' efforts in rearranging the subsections. The overall flow has improved significantly, making the presentation more natural and providing the reader with additional context where needed.

The authors have added a brief introduction to this section, which was missing in the previous version of the manuscript. While the idea of including an

introduction is useful, the current text does not effectively set up the content of Section 2. It reads more like a condensed abstract, lacking a clear overview of the topics covered in this section and referencing material that is actually discussed in the following section, which may confuse the reader.

Lines 121-122, "We present ... distributed manner.": I suggest clarifying here that this section, specifically subsection 2.1, provides a detailed examination of the model's architecture and the rationale behind key design choices, since a general introduction to the model was already given in the previous section. Currently, lines 121-122 simply repeat a condensed version of lines 103-107 and do not clearly indicate what the reader should expect from this section. Additionally, it would be helpful to note that the input and output data of DRRAiNN will be discussed in this section, as these aspects are closely related to the model's architecture.

Lines 122-123, "We evaluate ... design choices.": This is not discussed in this section, but rather in section 3. I suggest to remove this sentence.

Lines 123-124, "We demonstrate ... Awareness System.": The actual demonstration of model performance is part of section 3 "Results". In this section the study area, experimental setup, benchmark model, and evaluation metrics are introduced, and this can be mentioned here in the introduction of the section. They are essentially all components required to assess performance, but the results themselves are presented in the following section.

Line 124-126, "DRRAiNN achieves ... modeled dynamics.": This is a summary of your results and should not be mentioned in the methodology.

Thank you very much for these comments, which thoroughly discuss the introducing paragraph of the method section. We agree that this paragraph feels out of place and revised it accordingly.

Lines 131-132, "... a grid that spans the whole catchment area of the river network.": Some readers might mistakenly assume the static maps and meteorological forcings span only the DEM-delineated catchment area, which is common practice. However, one of the main motivations for developing DRRAiNN is that the effective catchment may extend beyond the DEM-delineated area, and DRRAiNN therefore also takes a larger domain as input. I suggest clarifying that the input data covers a larger domain that includes the DEM- delineated catchment area, but also extends beyond it.

We agree and now explicitly state that the grid spans a domain that is larger than the DEM-delineated area.

Lines 141-142: I appreciate the authors clear description of the nature of the input, internal states, and output of DRRAiNN.

Lines 148-149, "Despite being ... self-organizing nature.": I appreciate this important clarification, as it resolves a point that was unclear to me in the original version of the manuscript.

Figure 1: I appreciate the improvements made to this figure, which now provides a clear and solid understanding of the inner workings of the model, especially in combination with the text. My only remaining concern is the two-sided arrow beneath the box labelled "StationGRU," whose meaning is not yet clear.

Thank you for pointing this out, we clarified the meaning of the arrows in the caption as well as in the main text.

The description of the rainfall-runoff model (subsection 2.1.1) is somewhat difficult to follow, though this is understandable given the model's complexity, and I appreciate the authors' effort to explain it as clearly as possible. In contrast, the description of the discharge model (subsection 2.1.2) is clear, concise, and highly effective

Lines 195-196: I find this very interesting; I was already wondering why one would only aggregate embedded runoff at the stations and not at every upstream river grid cell.

I appreciate the thorough and detailed description of the data used in the study, provided in subsection 2.2. The data preprocessing, as well as the temporal and spatial resolution, are clearly reported, addressing one of my concerns with the previous version of the manuscript.

Line 257: Although I appreciate the authors' description of the temperature data, the extreme values convey limited information about local climate conditions. I suggest replacing these with, for example, the mean summer and winter temperatures.

We agree and replaced the values accordingly.

Figure 3 has improved considerably, and the issues I noted with this figure in the previous version of the manuscript have been adequately addressed.

Line 259: I highly appreciate the authors' description of the discharge characteristics.

Lines 281-284: I appreciate the authors pointing this out, and I agree that it is entirely reasonable to leave the evaluation of DRRAiNN under forecast-based conditions for future work.

Line 292: Changing the batch size during training, to counteract the increased memory use by the simultaneously changing truncation length is a very elegant

solution! I appreciate the authors highlighting this aspect of their work.

Thank you!

Line 293: I appreciate the authors mentioning the computational time for a forward pass, as this was not mentioned in the previous version of the manuscript.

Line 300-304: Reading between the lines, I understand that a CNN trained only on the original data may perform worse when presented with a rotated or reflected version of the data. To address this and improve the CNN's generalization, I understand the authors train it on various symmetries of the data. If this interpretation is correct, I suggest adding a brief sentence explaining the issue of data symmetries in CNNs. Additionally, although it is somewhat mentioned in line 304, it may be helpful to more explicitly state that GNNs do not suffer from this problem.

Thank you for pointing that out. We adjusted the paragraph accordingly, which will likely improve understanding.

Lines 315-326: I appreciate the authors' detailed explanation of the differences between EFAS and DRRAiNN, not only in terms of the models themselves but also their inputs, outputs, resolutions, and use cases. I also value their acknowledgment that the comparison of DRRAiNNperformance with EFAS is not entirely valid. However, as the authors clearly state, this is not their intention; their goal is not to outperform EFAS, but to provide a baseline comparison. In my opinion, their reasoning is entirely fair and justifies the comparison made.

Subsection 2.6: I appreciate the authors providing their reasoning for including each of these metrics, as I previously suggested reducing the number of metrics. With this explanation, I understand their rationale for using four different metrics.

Lines 356-357: The NSE does include the bias, but "in normalized form scaled by the standard deviation" of the target variable (Gupta et al., 2009, see https://doi.org/10.1016/j.jhydrol.2009.08. That was one of the main motivations to develop the KGE, which incorporates the correlation, bias, and variability independent from each other.

Absolutely correct, thank you! We fixed this accordingly.

**3. Results**

The reporting of results is clear and thorough, and the figures are well-designed. Especially Figure 6 and Figure 7 have much improved compared to the previous version of the manuscript. I also appreciate the thoughtful interpretation of

the results and model performance, including the explanation for the varying performance of the DRRAiNN model across different seeds and lead times.

My only significant issue with this section, which I also mentioned in my previous review, is the sometimes seemingly conflicting assessment of the attribution maps. I agree that, to a large extent, the attribution maps from DRRAiNN are expected to overlap with the DEM- delineated catchment areas and could thus be used as an indicator of physical plausability. However, as the authors themselves point out, subsurface flow can, in some cases, transcend DEM-delineated catchment boundaries.

Therefore, I suggest that the authors exercise caution when classifying a DR-RAINN model as better solely because its attribution maps show more overlap with traditional catchment delineations than another DRRAINN model (lines 461-464). To a certain extent, overlap between the DRRAINN attribution map and the DEM-derived catchment area is expected and is indeed an indicator of physical plausibility. However, beyond a certain point, increased overlap cannot be assumed to correlate with higher physical plausibility.

In case an attribution map is wildly different from the DEM-delineated catchment, it should indeed be assessed lower than one with more overlap. However, once the differences between two DRRAiNN models are small, caution is needed. As the authors note, DRRAiNN may be capable of detecting unobserved subsurface flow paths, though, as they also emphasize, further research is required to confirm this.

I understand the delicate balance between these potentially contradictory considerations, but I urge the authors to avoid rigorously stating that more overlap automatically indicates a better model, especially since they acknowledge that the DEM-delineated catchment map may not capture all flow paths. When the authors state that one DRRAiNN model instance has more overlap than another and should therefore be regarded as having more physical plausibility (lines 461-464), I suspect they are referring to cases where the lack of overlap from the latter model cannot be explained by subsurface flow alone. If that is the case, I suggest clarifying this for the reader.

If the lower overlap could also be due to undetected subsurface flow paths, I suggest the authors refrain from jumping to conclusions and remain open to the possibility that DRRAiNN may have identified such paths (as, for example, noted in lines 426–427).

Thank you for pointing this out again. We agree that our assessment still was not quite right. We now softened our conclusions and adjusted the paragraph accordingly. We also adjusted the paragraph about the ablation studies accordingly.

Lines 404-405: The quality of the observational data at these locations may influence the performance of both models. The authors are in a better position to assess this. If they consider this a potential reason for the reduced model performance at these locations, I suggest mentioning it here

We agree and added data quality to the list.

Lines 410-411: The NSE does include bias and variability, see Gupta et al. (2009, https://doi.org/10.1016/j.jhydrol.2009.08.003).

Again, thank you. Since a direct comparison of KGE and NSE is non-trivial and tells more about the metrics themselves than the stations, we removed this sentence.

Line 415: I assume the authors intended to indicate that darker areas correspond to regions of higher importance.

Yes, indeed. Thank you!

I appreciate the extensive series of ablations conducted by the authors.

**4. Discussion**

The discussion is well written, providing a clear summary of the work as well as several interesting suggestions for future research. I appreciate the authors' efforts in condensing the discussion.

Thank you so much!

**5. Conclusion**

The conclusion is well written and concise, and it clearly highlights the added value of the study.

Thank you!

**Technical corrections**

Line 39: A period is missing after "biases".

**List of relevant changes made in the manuscript**

- 1. Mention that we predict at gauging stations in abstract
- 2. State that evapotranspiration is difficult to measure (in contrast to impossible)
- 3. Mention saliency maps instead of integrated maps in introduction, as this is the technique we used
- 4. Improve transition from introduction to method chapter
- 5. Explicitly state that our domain in larger than the elevation-delineated catchment area
- 6. Describe the two arrows between Segment- and StationGRUs
- 7. Add threshold that was used to remove stations if coordinate correction is too large
- 8. Add mean temperature of summer vs winter instead of min and max temperatures
- 9. Briefly explain why CNNs benefit from data augmentation while GNNs don't need it
- 10. Correctly handle differences in KGE and NSE metrics
- 11. Explicitly state that lower Wasserstein distance could be due to correctly inferred underground flows
- 12. Add data quality as possible explanation for performance variance across stations